# Damage-responsive, maturity-silenced enhancers regulate multiple genes that direct regeneration in *Drosophila*

**Robin E Harris[1]\*, Michael J Stinchfield[1], Spencer L Nystrom[2], Daniel J McKay[2], Iswar K Hariharan[3]\***

[1]Arizona State University, Tempe, United States; [2]University of North Carolina at Chapel Hill, Chapel Hill, United States; [3]University of California, Berkeley, Berkeley, United States

**Abstract** Like tissues of many organisms, *Drosophila* imaginal discs lose the ability to regenerate as they mature. This loss of regenerative capacity coincides with reduced damage-responsive expression of multiple genes needed for regeneration. We previously showed that two such genes, *wg* and *Wnt6*, are regulated by a single damage-responsive enhancer that becomes progressively inactivated via Polycomb-mediated silencing as discs mature (Harris et al., 2016). Here we explore the generality of this mechanism and identify additional damage-responsive, maturity-silenced (DRMS) enhancers, some near genes known to be required for regeneration such as *Mmp1*, and others near genes that we now show function in regeneration. Using a novel GAL4-independent ablation system we characterize two DRMS-associated genes, *apontic* (*apt*), which curtails regeneration and CG9752/*asperous* (*aspr*), which promotes it. This mechanism of suppressing regeneration by silencing damage-responsive enhancers at multiple loci can be partially overcome by reducing activity of the chromatin regulator *extra sex combs* (*esc*).

**\*For correspondence:**
Robin.Harris@asu.edu (REH);
ikh@berkeley.edu (IKH)

**Competing interests:** The authors declare that no competing interests exist.

## Introduction

Tissue regeneration is a complex phenomenon that occurs in diverse taxa, and can result from a variety of mechanisms, including amplification of stem cells, changes in mature tissue identity, and de-differentiation and remodeling of established tissue (*Tanaka and Reddien, 2011*). Following tissue damage or loss, these processes promote the restoration of tissue size, structure and patterning, and are governed by coordinated programs of gene expression. However, in many organisms, regenerative capacity declines as an organism matures through development (*Yun, 2015*). The hind limbs of the anuran amphibian *Xenopus laevis* progressively lose the ability to recover from amputation as the tadpole develops through juvenile stages (*Dent, 1962*; *Overton, 1963*; *Muneoka et al., 1986*; *Wolfe et al., 2000*). Damaged cardiac tissue can completely regenerate in newborn mice, while the same injury inflicted just 7 days later results in fibrosis and scarring (*Porrello et al., 2011*; *Porrello et al., 2013*). This striking loss of regenerative capacity with increasing maturity is observed in diverse tissues of mammals (*Reginelli et al., 1995*; *Porrello et al., 2011*; *Cox et al., 2014*) including humans (*Illingworth, 1974*; *King, 1979*), and also in amphibians (*Dent, 1962*; *Freeman, 1963*; *Beck et al., 2003*; *Slack et al., 2004*) and invertebrates (*Smith-Bolton et al., 2009*; *Halme et al., 2010*; *Harris et al., 2016*). Remarkably, many of these same tissues continue their program of developmental growth even after they lose the ability to regenerate. As such, how a regeneration program becomes curtailed, and how this occurs independently of developmental growth, has yet to be elucidated.

The ability of *Drosophila* imaginal discs – the larval primordia of adult structures such as the wing and eye – to regenerate was originally explored via classic transplantation studies (*Ursprung and*

*Hadorn, 1962*). More recently, genetic methods in which the discs are damaged *in situ* by the temporally and spatially limited expression of pro-apoptotic genes have enabled larger-scale experiments in which the domain of tissue ablation can be regulated more precisely (*Smith-Bolton et al., 2009*; *Bergantiños et al., 2010*). Using these and other approaches, it was shown that imaginal discs readily regenerate at the beginning of the third larval instar (L3), but lose this ability over the course of L3 (*Smith-Bolton et al., 2009*; *Halme et al., 2010*; *Harris et al., 2016*). Multiple genes known to be upregulated in response to damage show less robust expression in more mature discs, which correlates with the loss of regenerative capacity. Recently it was shown that genome-wide changes in chromatin accessibility are associated with regeneration following genetically-induced cell death in wing discs (*Vizcaya-Molina et al., 2018*). However, these investigations were performed on discs at a single developmental stage when they still possessed high regenerative capacity, and therefore it remains to be seen how damage-induced changes to the epigenetic landscape might be altered in mature discs that have lost the ability to regenerate.

Wnt proteins play an important role in orchestrating regeneration in many organisms (*Stoick-Cooper et al., 2007*). Using a genetic ablation system, we previously investigated the progressive decrease in damage-responsive *wingless* (*wg*) expression in wing-imaginal discs as they mature. Following damage, *wg* expression requires a damage-responsive enhancer, BRV118, located between *wg* and *Wnt6* (*Schubiger et al., 2010*; *Harris et al., 2016*). We showed that this enhancer contains a damage-responsive module (BRV-B) containing multiple binding sites for the JNK-responsive transcription factor AP-1 and that these sites are essential for its damage-responsive activity. An adjacent and separate element, BRV-C, has no enhancer activity on its own, but can silence the damage-responsive expression mediated by BRV-B in cis in a maturity-dependent manner by promoting Polycomb-mediated silencing of the enhancer, characterized by highly localized H3K27 trimethylation. This localized epigenetic change, which spares more distant developmentally-regulated enhancers at the *wg*/*Wnt6* locus, provides a mechanism for selectively shutting off damage-responsive expression while preserving the ability of those genes to be expressed for normal development. Importantly, restoring *wg* expression in late L3 either by CRISPR/Cas9-mediated excision of the silencing element, BRV-C, or by expression of *wg* did not restore regeneration. This raises the possibility that multiple genes necessary for regeneration could be regulated similarly by damage-responsive enhancers that are also silenced in maturing tissues.

Using a genome-wide ATAC-seq approach as a guide, we have now identified additional damage-responsive enhancers that are silenced as larvae mature. Using a GAL4-independent tissue ablation system that we have developed, we show that genes associated with these elements are necessary for robust regeneration, thus demonstrating that the silencing of multiple such enhancers could account for the decrease in regenerative capacity as tissues mature. Proximity to such enhancers has also allowed us to identify novel regulators of regeneration. Finally, we show that modulating the activity of a specific chromatin regulator that alleviates silencing at such enhancers can promote regeneration in mature discs.

## Results

### A damage-responsive and maturity-silenced enhancer is also present at the *Mmp1* locus

To investigate the possibility that genes other than *wg* and *Wnt6* might be regulated by damage-responsive and maturity-silenced (DRMS) enhancers, we searched for modules with a similar bipartite organization to BRV118, the enhancer identified at the *wg*/*Wnt6* locus. The damage-responsive module of BRV118, BRV-B, contains multiple AP-1 binding sites that are essential for its ability to respond to tissue damage, and which are also found in the corresponding enhancer regions of other *Drosophila* species (*Figure 1A*). We have previously shown that multiple elements in the module required for silencing the enhancer in mature discs (BRV-C) are necessary (*Harris et al., 2016*) and that Polycomb group (PcG) proteins are necessary for the silencing activity of BRV-C. Indeed, a binding site of the PcG DNA binding factor Pleiohomeotic (Pho) (*Mohd-Sarip et al., 2002*) is present in BRV-C (*Figure 1A*). To identify other functional motifs, we compared the sequences from four highly

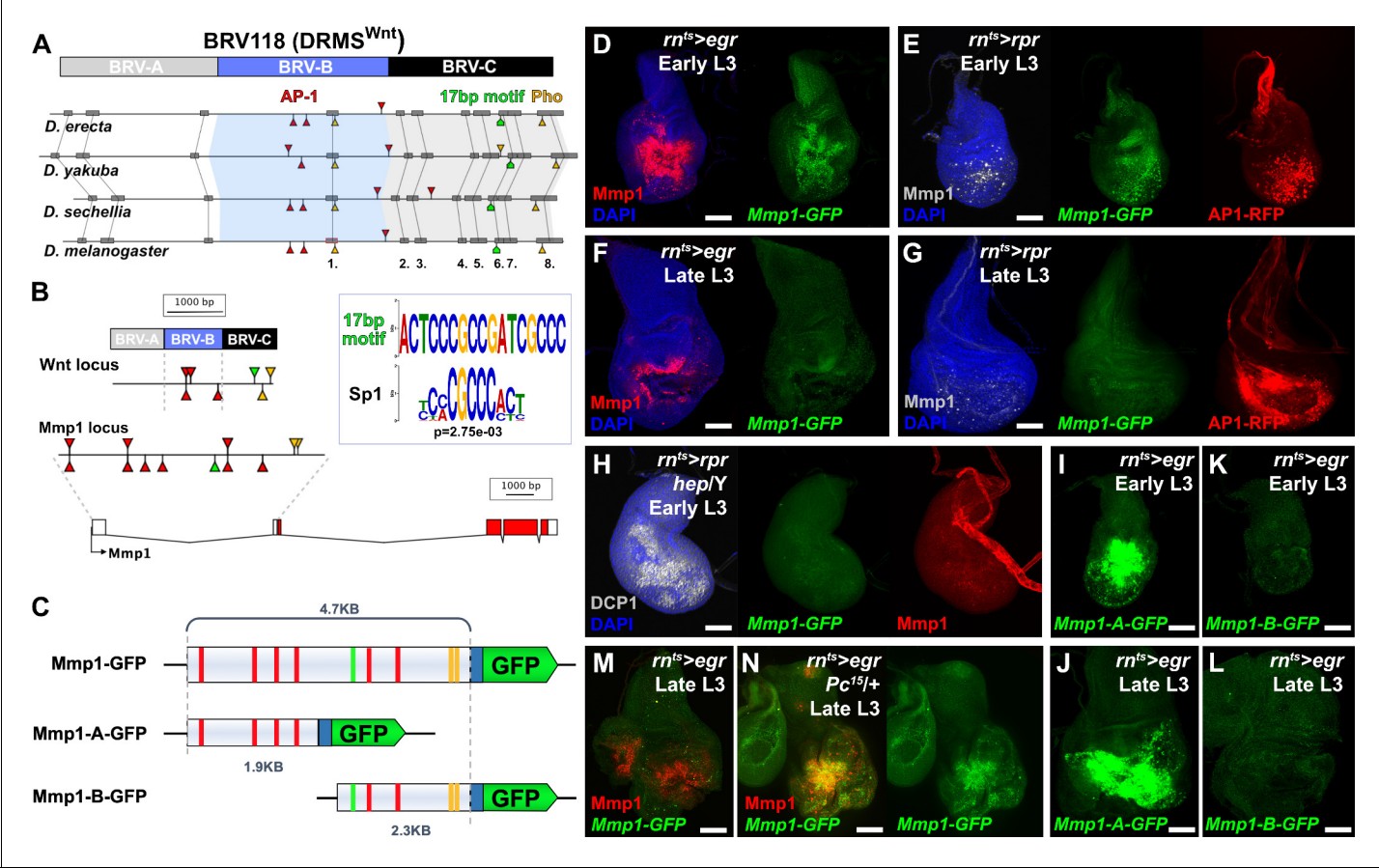

**Figure 1.** *Mmp1* is regulated by a bipartite damage-responsive enhancer with organizational similarity to BRV118 (DRMS^Wnt). (**A**) Schematic illustrating the conservation of the BRV118 enhancer at the Wnt locus (top) in four *Drosophila* species. The main damage-responsive region, BRV-B (blue box), the maturity silencing region, BRV-C, (black box) and their equivalent sequences in other species are indicated. Matching sequence of 50 bp or greater are shown as gray boxes and numbered. Also indicated are AP-1 binding sites (red arrowheads), Pleiohomeotic (Pho) sites (yellow arrowheads) and the conserved 17 bp motif (green markers). Binding site orientation is indicated by appearance above or below the line, (**B**) Schematic comparing the BRV118 enhancer from the Wnt locus (top) with that of a putative enhancer at the *Mmp1* locus (bottom). AP-1 and Pho binding sites, and the 17 bp motif consensus, are illustrated in both DNA sequences, as in (A). The 17 bp motif and the Sp1 motifs are shown (inset), (**C**) Schematics of the *Mmp1*-GFP and related reporters. AP-1 binding sites (red bars), Pho binding sites (yellow bars) and the 17 bp motif (green bars) are indicated. Blue box: *hsp70* minimal promoter, (**D**) Early L3 wing imaginal disc following ablation with *rn^ts>egr* stained for Mmp1 (red) and DAPI (blue), and showing activity of the *Mmp1-GFP* reporter (green), (**E**) Early L3 wing disc following ablation with *rpr*, showing levels of Mmp1 (gray), the activity of the *Mmp1-GFP* reporter (green) and the *AP-1-RFP* reporter (red), (**F**) Late L3 wing disc following *egr* ablation, showing that both the damage-induced Mmp1 (red) and *Mmp1-GFP* reporter expression (green) is weaker than that of early L3 discs, (**G**) Late L3 wing disc ablated with *rpr* as in (E), showing expression of both *Mmp1* and the reporter are weaker in late L3 discs, while *AP-1-RFP* remains strongly activated on both days, DAPI: blue, (**H**) Early L3 hemizygous *hep^-* mutant wing disc following *rpr* ablation, showing that neither Mmp1 (red) or the *Mmp1-GFP* reporter (green) is activated despite damage, indicated by dead cells (DCP1, gray), (**I-J**) Early L3 (I) and late L3 (J) wing discs bearing the *Mmp1-A-GFP* reporter (green) following *egr* ablation. The reporter is strongly activated, even in mature discs, (**K-L**) Early L3 (K) and late L3 (L) wing discs bearing the *Mmp1-B-GFP* reporter (green), showing no activity following *egr* ablation, (**M-N**) Mmp1 protein (red) and *Mmp1-GFP* expression (green) in late L3 *egr* ablated discs in a wild type (M) or *Pc^15* heterozygous mutant background (N), showing increased levels of Mmp1 and GFP with reduced *Pc* gene function. Scale bars = 50 μm.

The online version of this article includes the following figure supplement(s) for figure 1:

**Figure supplement 1.** The *Mmp1* enhancer is activated by ectopic JNK signaling and physical wounding.

**Figure supplement 2.** The *Mmp1* enhancer is damage activated and is defined by a 1 kb fragment.

---

related *Drosophila* species and found several conserved regions (*Figure 1A*). Screening for stretches of identical DNA sequence 50 bp or larger, we identified a single region (region 1) within BRV-B that is close to the three AP-1 binding sites that we previously showed are required for damage-responsive expression (*Harris et al., 2016*). By comparison, the BRV-C module has multiple completely-conserved regions, including two regions over 100 bp in length (regions 6/7 and 8), one of which

(region 8) contains the conserved Pho binding site (*Zhu et al., 2011*). By conducting BLAST searches of the genome using sequences from these highly conserved regions, we found an exact copy of a 17 bp sequence from region 6/7 present within an enhancer previously identified in the *Matrix metalloproteinase 1* (*Mmp1*) locus (*Figure 1B*, *Uhlirova and Bohmann, 2006*). We compared this 17 bp sequence with a library of known *Drosophila* transcription factor binding sites using a motif comparison tool (*Gupta et al., 2007*) and found that part of this sequence matches an Sp1 binding site (*Figure 1B*). Sp1 binding sites are known to be required for the activity of Polycomb Response Elements (PREs) and have been identified in most molecularly-characterized PREs (*Brown et al., 2005*), while an Sp1 family member, Ssps, has been shown to bind different PREs and contribute to silencing (*Brown et al., 2005*; *Brown and Kassis, 2010*). In the ~5 kb surrounding this motif at the *Mmp1* locus are two conserved Pho binding sites and six AP-1 binding site (*Figure 1B*).

To investigate this region, 4.7 KB of DNA upstream of the *Mmp1* coding sequence, which includes the 17 bp motif, and the AP-1 and Pho binding sites, was cloned upstream of a minimal promoter and GFP coding sequence. This transgenic reporter, *Mmp1-GFP* (*Figure 1C*) showed little activity during normal development in undamaged discs, only weakly recapitulating the *Mmp1* expression that normally occurs in the developing air sac in late L3 discs (*Wang et al., 2010*; *Figure 1—figure supplement 1A*). In contrast, upon genetic ablation in early L3 discs (day 7) using *rn-GAL4*, *GAL80^{ts}*, *UAS-eiger* (hereafter *rn^{ts}>egr*), the reporter is strongly activated in a pattern that resembles endogenous damage-induced Mmp1 protein (*Figure 1D*). A similar pattern of expression, albeit weaker, is observed when *reaper* (*rpr*) is used instead of *egr* to kill cells, and this expression is coincident with an AP-1 reporter (*Figure 1E*). Ablation in the absence of JNK activity using a *hep* mutant background (*hep/Y*) fails to induce the reporter or *Mmp1* (*Figure 1H*). Thus, JNK signaling is a necessary input into the enhancer, as it is for damage-induced *Mmp1* expression. Conversely, ectopic activation of JNK signaling through expression of *hep^{CA}* leads to strong reporter activation (*Figure 1—figure supplement 1B*). Physical wounding of these discs followed by *ex vivo* culture also results in activation of the *Mmp1* reporter at the wound edge, coincident with *Mmp1* expression (*Figure 1—figure supplement 1C*). Consistently, the *Mmp1-GFP* reporter also recapitulates the weaker expression of *Mmp1* in response to genetic ablation with either *egr* or *rpr* in late L3 discs (day 9, *Figure 1F–G*), despite a robust level of JNK activity, as indicated by the AP-1 reporter (*Figure 1G*). Together these data indicate that this region of the *Mmp1* locus contains an enhancer that is both damage-responsive and maturity-silenced (DRMS).

To directly test whether this DRMS enhancer has separable damage-activated and maturity-silencing elements, two reporter lines, *Mmp1-A-GFP* and *Mmp1-B-GFP*, were generated using enhancer fragments (*Figure 1C*) and inserted into the same transgene landing site as the original *Mmp1-GFP* to make their activity directly comparable. *Mmp1-A* was strongly activated specifically in response to damage, more so than the full-length enhancer (*Figure 1I–J*). Moreover, in the absence of the *Mmp1-B* sequences it can be activated equally as strongly in both early and late L3 discs (*Figure 1I–J*). In contrast, *Mmp1-B*, which contains the 17 bp motif and conserved Pho binding site yielded no enhancer activity in ablated young or old discs (*Figure 1K–L*), despite containing two predicted AP-1 binding sites (*Figure 1C*). Further subdivision of the *Mmp1-A* region showed that the majority of damage-responsive expression is driven by a ~ 1 kb section of DNA bearing three high consensus AP-1 binding sites (*Figure 1—figure supplement 2A–C*). None of the generated reporters showed significant expression in the absence of ablation (*Figure 1—figure supplement 2E–I*). We also examined the activity of the *Mmp1-GFP* reporter in a PcG mutant background, which we previously showed de-repressed the BRV118 enhancer in the *wg/Wnt6* locus (hereafter DRMS^{Wnt}) in older damaged discs compared to wild type (*Harris et al., 2016*). Expression of *Mmp1-GFP* and Mmp1 protein in a *pc^{15}/+* ablated late L3 disc was significantly stronger compared to the wild type control (*Figure 1M–N*), indicating that Polycomb-mediated epigenetic silencing is necessary to limit *Mmp1-GFP* activation in mature tissues. Thus, the functional organization of this enhancer is very similar to the one we have characterized at the *wg/Wnt6* locus, with clearly separable damage-responsive modules and silencing modules. Also, like DRMS^{Wnt}, its age-dependent silencing is dependent upon PcG function. Due to the similarity of organization with DRMS^{Wnt}, we will henceforth refer to this region as DRMS^{Mmp1}. Thus, we have shown that another gene that has been demonstrated to function in both growth and tissue remodeling is regulated by a DRMS enhancer.

## Genome-wide identification of DRMS enhancers using ATAC-seq

While DRMS$^{Mmp1}$ and DRMS$^{Wnt}$ share sequence motifs, it is possible and even likely that other such enhancers rely on different mechanisms to be activated or silenced, and thus potentially lack these motifs. To search for such enhancers without sequence bias, we employed ATAC-seq, a genome-wide assay of chromatin accessibility (*Buenrostro et al., 2015*). We compared chromatin profiles of *rn$^{ts}$>egr* ablated wing discs from both early and late L3 larvae (reflecting times of high and low regenerative capacity, respectively), as well as from identically staged unablated discs (*Figure 2—figure supplement 1A*). We performed three full biological repeats for each condition yielding a total of 14,142 open chromatin peaks after merging overlapping peaks from each condition (see Materials and methods for full details of quality control and data analysis parameters). A Pearson correlation analysis on CPM-normalized counts from DEseq2 shows high correlation between replicates (r >0.9) and demonstrates that the data cluster first by developmental stage and then by whether or not the tissue had been damaged (*Figure 2—figure supplement 1B*). When analyzing the chromatin accessibility data, we were surprised to find that only a small number of peaks in each pairwise comparison met with a standard statistical cutoff (padj <0.05), particularly when comparing chromatin profiles of damaged and undamaged discs (151 altered in early L3 with 90 becoming more accessible, 27 altered in late L3 with 16 becoming more accessible, *Supplementary file 1*). We speculate that this is likely due to the use of whole discs in our protocol, as signals from regenerating cells only comprise a small portion of the disc and are potentially diluted by the chromatin profile of cells of undamaged tissue. To overcome this limitation, we used less stringent statistical limits, analyzing peaks with a padj <0.1 and log2FC >0.5 (*Supplementary file 2*). Using these cutoffs, we observe the same trend but now identify 222 regions that become more open upon damage in early L3 compared to undamaged controls, and 33 that become more accessible in late L3 discs (*Figure 2—figure supplement 1C–E* and *Supplementary file 2*). Of these, 12 regions consistently open upon damage at both time points. Thus, the chromatin landscape, at least as assessed by this criterion, is more responsive to damage in immature than mature discs, which in principle could contribute to the reduction in regenerative capacity. When comparing damaged discs at the different developmental timepoints, there are 729 regions that are significantly less accessible in damaged late L3 discs versus damaged early L3, and therefore become silenced with maturity in the context of damage (*Figure 2—figure supplement 1C and E*, and *Supplementary file 2*). We find that 28 (13.3%) of the 222 early L3 damage-responsive are included in this category, thus identifying them as putative damage-responsive and maturity-silenced (DRMS) regions (*Figure 2—figure supplement 1C* and *Supplementary file 2*). Assaying the position of the 28 DRMS peaks relative to the total open chromatin detected by ATAC-seq shows they are enriched at genomic locations categorized as actively transcribed introns and other open chromatin, while being relatively reduced at actively transcribed promoters and exons (*Figure 2—figure supplement 1F*; *Kharchenko et al., 2011*), consistent with these peaks localizing to enhancers.

Visualizing the chromatin accessibility at the experimentally validated *wg/Wnt6* and *Mmp1* enhancer loci shows that both regions become more accessible upon damage in early L3 discs, but not in late L3, consistent with DRMS behavior (*Figure 2A–B*). However, despite the less stringent statistical cutoffs, the peaks representing these enhancers are not detected as damage-responsive, while only the Wnt enhancer is designated as maturity-silenced (*Supplementary file 2*). This is likely because damage-specific changes in chromatin accessibility at these enhancers within regenerating cells are diluted by surrounding cells that comprise the rest of the disc. This is particularly evident at the DRMS$^{Mmp1}$ enhancer, which maintains significant accessibility regardless of damage or age (*Figure 2B*). Due to this limitation, we have used the statistical analysis simply as a guide alongside manual curation of browser traces to identify potential DRMS enhancers that could then be tested experimentally.

With this approach, we identified an additional region close to and upstream of the characterized DRMS$^{Mmp1}$ region that also showed characteristic DRMS behavior enhancer and was identified as a maturity-silenced region (*Figure 2B* and *Supplementary file 2*). We generated a GFP reporter to the region spanning this region (*Mmp1-US-GFP*) and observed damage-responsive, maturity-silenced expression correlating with its accessibility indicated by ATAC-seq (*Figure 2C–C'*). As for the *Mmp1-GFP* reporter, *Mmp1-US-GFP* showed no activity in the absence of damage (*Figure 1—figure supplement 2D*). These results show that regenerative *Mmp1* expression is likely regulated

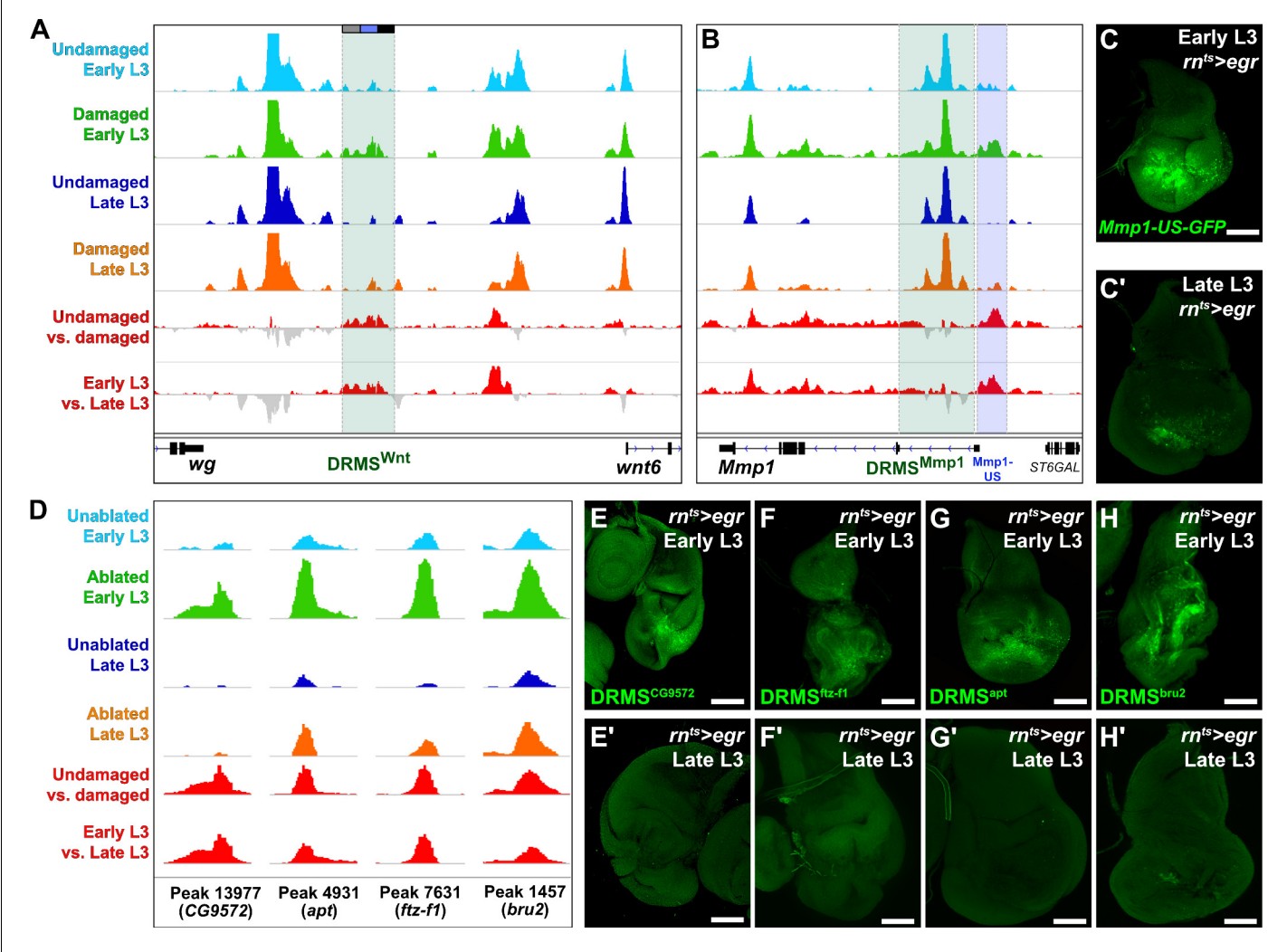

**Figure 2.** ATAC-seq of regenerating discs identifies damage-responsive and maturity-silenced regions. (A–B) ATAC-seq chromatin accessibility trace (z scores) at the Wnt (*wg/Wnt6*) locus (A) and the *Mmp1* locus (B) for the four conditions indicated. Traces showing the difference between early L3 undamaged versus damaged discs (damage-responsiveness), and early L3 damaged versus late L3 damaged discs (maturity silencing) is shown in the bottom two traces (subtracted z scores, red and gray). The dark green boxes indicate the experimentally validated DRMS^Wnt enhancer (BRV118, [*Harris et al., 2016*]) and DRMS^Mmp1 enhancer (this work), the blue box indicates the *Mmp1* upstream region (*Mmp1-US*) tested for DRMS activity, (C-C') Characterization of the *Mmp1-US* enhancer. Early L3 (C) and late L3 (C') *rn^ts>egr* ablated discs bearing GFP reporter of the *Mmp1-US* region (peak 5105), showing damage-responsive and maturity-silenced behavior, (D) ATAC-seq chromatin accessibility traces of the peaks (and associated genes) indicated, chosen for *in vivo* validation based on their strong DRMS signatures, (E-H') Early L3 (E-H) and late L3 (E'-H') *rn^ts>egr* ablated discs bearing GFP reporters of the peaks indicated in (D). Each reporter has damage-responsive expression in early L3 discs, which is reduced in damaged late L3 discs, consistent with chromatin accessibility of the region. Scale bars = 50 µm.

The online version of this article includes the following figure supplement(s) for figure 2:

**Figure supplement 1.** ATAC-seq identifies damage-responsive chromatin changes that are located at genomic regions resembling enhancers.

**Figure supplement 2.** *In vivo* validated DRMS enhancers are not activated in the absence of damage.

via multiple enhancers, and that our approach can be successful in identifying actual enhancers that are both damage-responsive and maturity-silenced.

To identify genes regulated by the putative enhancer regions we identified the closest two genes for each of the 222 early L3 damage-responsive peaks and the 28 DRMS peaks (nearest two upstream or downstream genes regardless of orientation, *Supplementary file 3*) since the majority of known enhancers are thought to regulate their immediately flanking genes (*Kvon et al., 2014*). Several genes known to be involved in the regeneration are adjacent to damage-responsive peaks,

including *kay*, *upd1* and *zfh2* (*La Fortezza et al., 2016*; *Supplementary file 3*). To directly assess whether some of the regions identified function as DRMS enhancers *in vivo*, we generated transgenic reporter lines of four candidate regions chosen based on the strength of their DRMS signature and their proximity to genes that were shown to be upregulated in an RNA-seq analysis of blastema cells in regenerating discs (*Khan et al., 2017*). ATAC-seq traces for regions near the genes *CG9572*, *ftz transcription factor 1* (*ftz-f1*), *apontic* (*apt*) and *bruno2* (*bru2*) are shown in *Figure 2D*. These were cloned upstream of a basal promoter driving GFP. All four reporters were tested under both ablated and unablated conditions in early and late L3 discs, and each showed damage-induced expression in early L3 discs of varying strength (*Figure 2E–H*). Importantly this expression was consistently reduced in late L3 ablated discs (*Figure 2E'–H'*), and none of the regions tested showed activity during development in undamaged L3 discs (*Figure 2—figure supplement 2A–D*). Thus, the DNA corresponding to these peaks are *bona fide* damage-responsive enhancers that are silenced with maturity, similar to the DRMS$^{Wnt}$ and DRMS$^{Mmp1}$ enhancers. These data demonstrate the utility of chromatin accessibility to identify cis-regulatory regions involved in shaping the transcriptional response to damage and show that genes expressed in blastema cells following injury (*CG9572*, *ftz-f1 apt*, and *bru2*) are adjacent to experimentally validated DRMS enhancers.

## A novel combinatorial expression system, DUAL Control, allows genetic manipulation of regenerating tissues

To test whether specific genes are indeed necessary for regeneration we sought to examine how manipulating their expression in damaged discs would influence tissue regrowth. The ablation system that we have been using thus far induces cell death by rendering GAL4 active for 40 hr at 30°C during L3, driving expression of a pro-apoptotic gene under UAS control. As such, other genes expressed under the control of UAS elements are also only active during the time of ablation and are restricted to the cells targeted for death. To overcome these limitations and take advantage of the extensive collections of UAS-driven RNAi lines and similar UAS-based tools, we developed a novel genetic ablation system that is independent of GAL4/UAS (*Figure 3A*). This system uses the bacterial transcriptional regulator LexA and its binding motif LexAOp, which have been used in *Drosophila* as an independent alternative to GAL4/UAS (*Lai and Lee, 2006*; *Pfeiffer et al., 2010*). Previous approaches have taken advantage of this alternative regulator to explore regeneration by using LexA fused to the GAL4 activator domain (LexA::GAD) to ablate the wing disc, while driving UAS-transgene expression using regular GAL4 in a spatially distinct domain (*Yagi et al., 2010*; *Santabárbara-Ruiz et al., 2015*; *Vizcaya-Molina et al., 2018*). However, as GAL80 is required to regulate both LexA::GAD and GAL4 in this arrangement, they are still temporally linked, limiting transgene expression solely to the period of ablation rather than during the subsequent phase of regeneration.

To overcome this problem, we have developed a system specifically for the purpose of manipulating gene expression in the regenerating tissue. First, to permit both temporal and spatial control of ablation in a GAL4-independent way, we generated a LexAOp-binding transcriptional activator that can be briefly and specifically activated in the wing disc by a heat shock (*Figure 3A*). Two transgenes were generated: the DNA binding domain (DBD) of LexA under the control of a *spalt* (*salm*) enhancer (*Jory et al., 2012*) and the transcriptional activator p65 domain (*Schmitz and Baeuerle, 1991*) under the control of the *hsp70* heat shock promoter. Each domain bears a complementary leucine zipper (*Ting et al., 2011*), which allows formation of the full chimeric LexA::p65 only in the cells that express both components. We combined these two transgenes with either *lexAOp-rpr* or *lexAOp-egr*, resulting in a system that permits tissue-specific ablation of the medial wing pouch in response to a heat shock, and which is entirely independent of GAL4/UAS (*Figure 3A*). Second, we developed a way of expressing GAL4 in the tissue surrounding the ablated region that is activated concurrently with the ablation. Since this system permits the simultaneous and independent use of both LexA and GAL4, each with separate spatial and temporal control, we named this system Duration And Location Control, or DUAL Control.

To drive UAS-transgene expression we established a DUAL Control stock that also includes a pouch-specific GAL4 under the control of either a *PDM2* or *DVE* enhancer (*Jory et al., 2012*), both of which target expression to regions of the wing pouch surrounding the region that would be ablated, i.e. the region that will include the regeneration blastema (*Figure 3C–D*). Although both enhancers drive in similar spatial patterns within the pouch, our experiments show that *DVE* has consistently stronger expression (data not shown). These GAL4 lines were generated with a flip-out

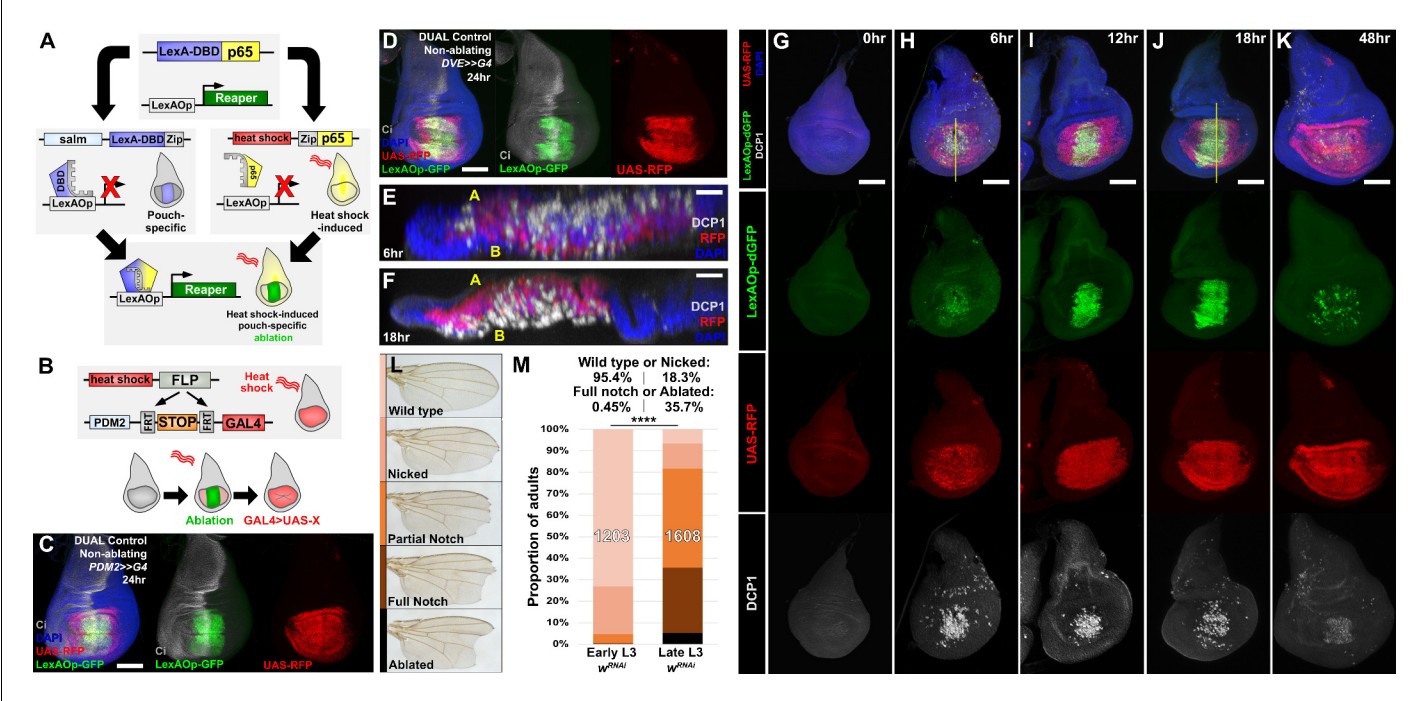

**Figure 3.** DUAL Control: a novel genetic ablation system to manipulate gene expression in regenerating discs. (A) Schematic of the DUAL Control genetic ablation system, based on split LexA. See manuscript text for details. *salm: spalt* enhancer, Zip: Leucine zipper domain, p65: transcriptional activator domain, (B) Schematic of the *PDM2>>GAL4* driver used in DUAL Control to manipulate gene expression in regenerating cells. Heat shock-induced FLP removes the transcriptional stop cassette through FRT recombination, allowing a single heat shock to activate both ablation (green) and GAL4 expression (red) in different cell populations, (C) Non-ablating DUAL Control with *PDM2>>GAL4* crossed to a double fluorescent tester stock, bearing *UAS-RFP* (red) and *lexAOp-GFP* (green), heat shocked at early L3 (day 3.5 at 25°C) and imaged 24 hr post heat shock (PHS). Cells that can be ablated (*salm* domain) are marked by GFP (green), while the surrounding cells that can be genetically manipulated (*PDM2* domain) are marked by RFP (red). The ablation domain straddles the compartment boundary, indicated by Ci staining (gray), (D) As in (C), using non-ablating DUAL Control with the stronger *DVE>>GAL4* driver. The expression of the *DVE* domain is similar to *PDM2*, (E-F) Sections through discs seen in (H) and (J), showing apoptotic cells (DCP1, gray) and pouch cells (RFP, red). The majority of dead cells and debris present within the disc proper at 6 hr PHS (E) is extruded basally from the disc epithelium by 18 hr PHS (F). A: Apical surface of the disc proper epithelium, B: Basal surface of the disc proper epithelium, DAPI, blue, (G-K) Time course of DUAL Control ablation with *rpr* on day 3.5 discs bearing *lexAOp-dGFP* (fast-degrading) and *UAS-RFP*, imaged at the indicated number of hours PHS. GFP (green), RFP (red) and cell death (DCP1, gray) can be detected at 6 hr PHS, and persist until 18 hr PHS. At 12 hr PHS the rate of new dead cell production decreases, while DCP1 positive cells and GFP label persist. GFP expression subsequently declines and is mostly absent by 48 hr PHS. GAL4 expression (RFP) is consistent from 6 hr PHS to pupariation throughout the regeneration period. DAPI, blue. Yellow lines in (H) and (J) indicate cross-sections in (E) and (F), (L) Classification of adult wing phenotypes following ablation and regeneration, (M) DUAL control used to ablate discs with *rpr* in early (day 3.5) or late (day 4.5) L3 discs, assayed for regeneration by wing size. Graphs illustrate the proportion of adults that eclose with the wing phenotypes indicated in (L), demonstrating the loss of regenerative capacity that occurs between early and late L3. In this and subsequent figures, the number of flies scored is labeled on the graph of each genotype, and the percentage of each genotype characterized as 'Wild type or Nicked' and 'Full notched or Ablated' is indicated above. *UAS-w^RNAi* is used as a control for regeneration scoring. **** p <0.0001, Fisher exact test. Scale bars in (E) and (F) = 20 μm, other scale bars = 50 μm.

The online version of this article includes the following figure supplement(s) for figure 3:

**Figure supplement 1.** Activation of DUAL Control with various heat shock durations.

**Figure supplement 2.** *egr*-induced ablation with DUAL Control is weaker than that of *rpr*.

cassette to permit heat shock-induced FLP-mediated activation (*PDM2>>GAL4* or *DVE>>GAL4*, *Figure 3B*). Thus, a single heat shock can simultaneously induce a pulse of LexA-driven ablation of the medial pouch and activate sustained GAL4 expression in the surrounding cells that contribute to the regenerated pouch (*Figure 3C–D*; *Herrera et al., 2013*; *Verghese and Su, 2016*). Since ablation occurs as a discrete pulse resulting from a heat shock rather than from the prolonged expression of GAL4 induced by a temperature change, this system has the advantage of clearly separating ablation from the period of regeneration. Similarly, as this system does not rely on temperature-sensitive

GAL80 to regulate either ablation or GAL4 expression, these experiments can be performed at higher temperatures to significantly reduce experimental duration.

To concurrently examine the temporal expression of lexAOp- and UAS-regulated transgenes in the absence of ablation, we crossed a non-ablating DUAL Control *DVE>>GAL4* stock to *lexAOp-dGFP; UAS-RFP* and heat shocked early L3 larvae (day 3.5 at 25°C). We found that both the LexA and the GAL4 components of the system can be activated with a single 37°C heat shock of duration as short as 10 min, but that a single 45 min heat shock was optimal (*Figure 3—figure supplement 1A–D*). We then examined a time course of ablation using *rpr* with DUAL Control *DVE>>GAL4* (*Figure 3G–K*) and found that cells expressing *rpr* (dGFP positive) can be detected within 6 hr post-heat shock (PHS, *Figure 3H*). At this time point the majority of the *salm* domain stains positively for activated caspase, which is mostly observed within the disc proper (*Figure 3E*). At 18 hr PHS the majority of cell corpses had been extruded basally from the disc (*Figure 3F and J*). At 24 hr the epithelium had mostly regained a normal appearance and by 48 hr PHS the associated debris was minimal, while regeneration was complete (*Figure 3K*). Despite using a fast-degrading GFP reporter (dGFP) with a half-life of only a few hours (*Lieber et al., 2011*), we found that fluorescence induced by LexA::p65 persisted in the disc after the initial heat shock, including in cells within the ablated *salm* domain (*Figure 3I–J*). This is likely due to these cells activating the *LexA-DBD* transgene as they take on distal pouch identity, which functions together with residual p65 to express *lexAOp-GFP*. These cells lack DCP1 staining however, suggesting that this level of LexA::p65 activity is enough to drive GFP expression but not ablation. By comparison, *UAS-RFP* expression resulting from heat shock induced *DVE>>GAL4* was observed at 6 hr PHS (*Figure 3H*) in the cells surrounding the ablated *salm* domain, and persisted throughout the recovery period (*Figure 3H–K*), demonstrating that cells that drive the regenerative growth can be targeted for manipulation using DUAL Control, and that ablation and UAS-driven gene expression can be temporally and spatially separated.

Using DUAL Control we examined the effect of inducing ablation at different developmental time-points. Larvae were heat shocked at early and late L3 stages (days 3.5 and 4.5 at 25°C) and the extent of regeneration was measured by assaying the size of the resulting adult wings (*Figure 3L–M*). The adult wings that develop from *rpr*-ablated discs display a series of phenotypes, which we categorized into discrete groups: 'wild type' for those indistinguishable from unablated wings (and therefore likely to be fully regenerated), 'nicked' for those with some margin loss, 'partial notch' or 'full notch' for those with significant loss of both margin and wing blade tissue, and 'ablated', describing those that had lost the entire distal wing (*Figure 3L*). When ablated in early L3, around 95% of wings produced were in the 'wild type' or 'nicked' category, indicating that regeneration was mostly complete (*Figure 3M*). By comparison, ablation in late L3 yielded many more 'full notch' or 'ablated' wings (*Figure 3M*), demonstrating the loss of regenerative capacity in wing discs during development. Ablation with *egr* resulted in weaker adult phenotypes (*Figure 3—figure supplement 2A*) and lower levels of activated caspase (*Figure 3—figure supplement 2B–E*). We therefore used ablation with *rpr* for subsequent wing scoring assays. RNAi targeting the *white* gene (*UAS-w^RNAi*) was used as a control for the other RNAi lines used in this work because of their similar genetic background and the lack of observable influence on disc regeneration.

## Manipulating the activity of characterized growth regulators using DUAL Control alters regeneration

To test our ability to interrogate factors that potentially influence regeneration we manipulated several previously identified regulators of regeneration using DUAL Control. Both Wg and Mmp1 protein are detected following ablation with *rpr* (*Figure 4—figure supplement 1A–B*), and more so with *egr* due to its stronger activation of the JNK pathway (*Figure 4A–B*). Both genes have experimentally validated DRMS enhancers, which are activated by DUAL Control ablation (*Figure 4A–B*). Targeting either *wg* or *Mmp1* for knockdown with RNAi using this system strongly decreases the damage-induced expression of both proteins, while the amount of cell death appears unaffected (*Figure 4C–D*). The extent of regeneration, as assessed by the change in adult wing phenotypes in response to ablation with *rpr*, is reduced in each case (*Figure 4E*). Knockdown of *Mmp1* for the same duration in the absence of ablation yields little to no effect on adult wings (*Figure 4—figure supplement 1C*), while knockdown of *wg* without ablation produces patterning defects localized to the distal wing edge (*Figure 4—figure supplement 1D*) that are clearly distinguishable from the wing tissue loss that follows ablation. Thus, *wg* is required for both regrowth following damage and

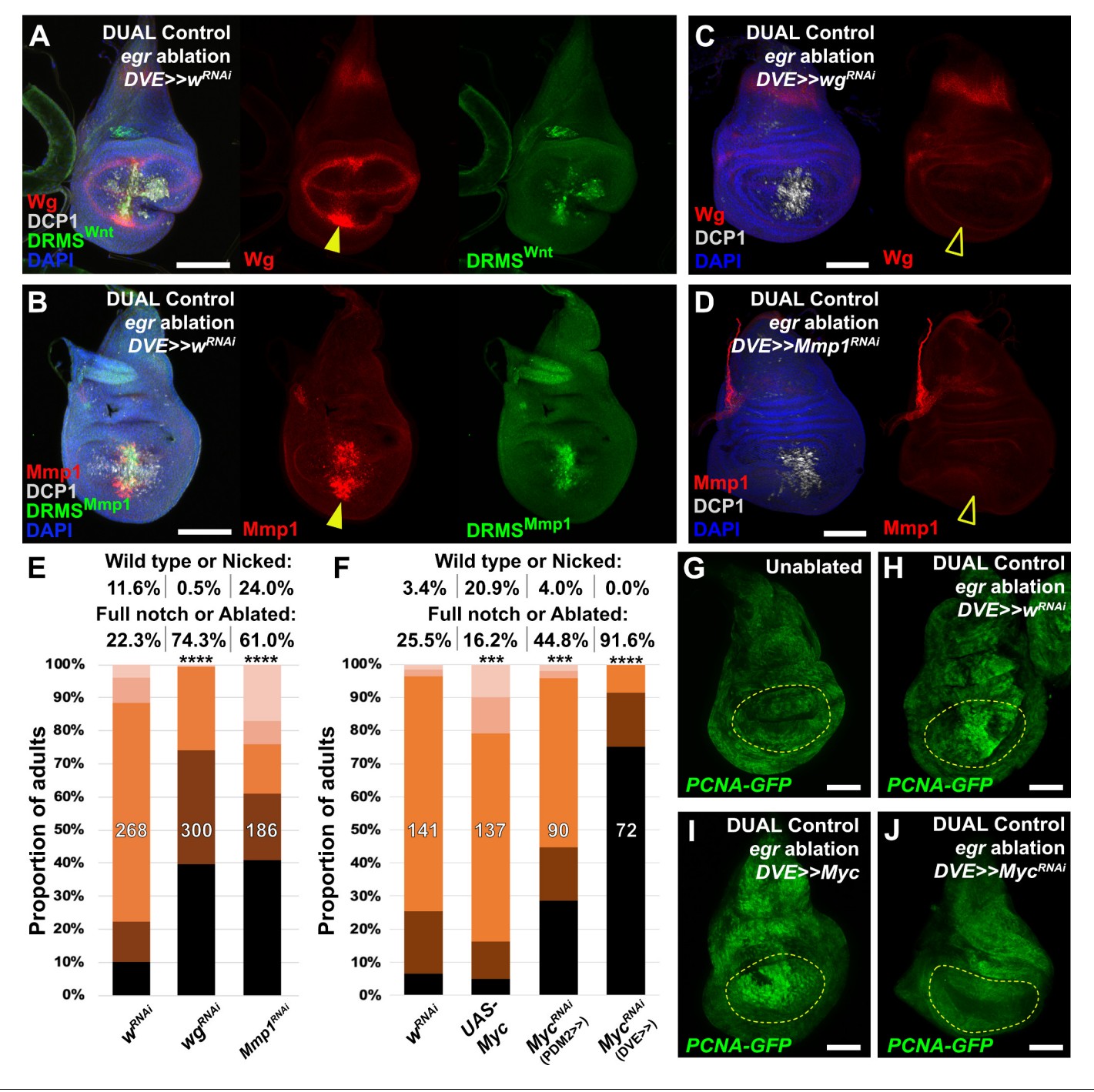

**Figure 4.** Manipulation of genes previously known to function during regeneration using DUAL Control. (A–B) Early L3 discs bearing the DRMS^Wnt^-GFP reporter (A) or the DRMS^Mmp1^-GFP reporter (B), ablated with *egr* using DUAL Control expressing a control RNAi (*UAS-wRNAi*) and imaged after 24 hr. Ablation activates expression of both *wg* and *Mmp1* (red, arrowheads), and both DRMS reporters (green), (C-D) RNAi knockdown of damage-induced *wg* (C) and *Mmp1* (D) expression (red) with *DVE>>GAL4* in DUAL Control *egr* ablated discs. Note that developmental expression of *wg* in the hinge and notum, and *Mmp1* in the tracheal tubes is unaffected by the knockdown, which is limited to the regenerating pouch tissue (open arrowheads). DCP1: gray, DAPI: Blue, (E) RNAi knockdown of DRMS-regulated genes *wg* and *Mmp1* with *DVE>>GAL4* in early L3 discs ablated with *rpr* using DUAL Control demonstrates their requirement for regeneration compared to *w^RNAi^* control. **** p <0.0001, Fisher exact test, (F) Manipulation of *Myc* alters regenerative ability of discs. Ectopic activation with *UAS-Myc* leads to improved wing sizes in late L3 *rpr* ablated discs compared to *w^RNAi^*, while knock down of *Myc* using *PDM2* or *DVE* drivers prevents regrowth, with the *DVE* driver having a stronger effect, *** p = 0.0006 (*UAS-Myc*), *** p = 0.0002 (*Myc^RNAi^*, *PDM2>>*), **** p <0.0001, Fisher exact test, (G-J) Discs bearing a *PCNA-GFP* reporter of E2F activity, and thus proliferation, in

*Figure 4 continued on next page*

*Figure 4 continued*

unablated discs (**G**), discs ablated with *egr* using DUAL Control (**H**), ablated with *egr* using DUAL Control and ectopically expressing *Myc* in the pouch with *DVE>>GAL4* (**I**), and ablated with *egr* using DUAL Control and expressing *Myc* RNAi in the pouch with *DVE>>GAL4* (**J**). Proliferation (GFP, green) is increased in blastema cells as a result of ablation, strongly upregulated throughout the pouch as a result of ectopic *Myc*, and significantly reduced in cells with *Myc* knockdown. Yellow outline indicates the pouch. Scale bars = 50 μm.

The online version of this article includes the following figure supplement(s) for figure 4:

**Figure supplement 1.** Characterizing known regulators of regeneration using DUAL Control.

repatterning. We also tested manipulation of *Myc*, a potent growth regulator shown to be sufficient to improve regeneration of late L3 discs (***Smith-Bolton et al., 2009***; ***Harris et al., 2016***). Knockdown of *Myc* using RNAi directed to regenerating tissue with either DUAL Control *PDM>>GAL4* or *DVE>>GAL4* following ablation with *rpr* showed a dramatic reduction in regeneration (***Figure 4F***). Consistently, overexpression of *Myc* improves adult wing size and morphology (***Figure 4F***). An E2F reporter (*PCNA-GFP*) shows that these phenotypes likely result from changes in damage-induced proliferation in response to altered levels of *Myc* (***Figure 4G–J***). JAK/STAT signaling is also an important regulator of regenerative growth, and we have identified damage-responsive peaks associated with *upd1* and *upd2* (***Supplementary file 3***). Knockdown of the JAK/STAT *upd1* or *upd2* appears to strongly reduce regeneration (***Figure 4—figure supplement 1E***), while knockdown of *upd3*, which does not have a damage-responsive peak, also mildly reduces regeneration but the effect is less significant (***Figure 4—figure supplement 1E***). Since the effects of the three *upd* genes could be additive on pathway function, we disrupted pathway components that are thought to function downstream of all three genes. Knockdown of the transcription factor *Stat92E* or receptor *domeless* (*dome*) also results in less-complete wings (***Figure 4—figure supplement 1E***), confirming the requirement for JAK/STAT signaling in regenerating tissue surrounding the ablation domain. Together, these data show that several genes either associated with experimentally validated DRMS enhancers or with damage-responsive peaks identified in this study are functionally necessary for regeneration, and demonstrate that DUAL Control is a powerful tool to manipulate gene function in regenerating tissue, circumventing the limitations of previous ablation systems.

## *asperous* (*CG9572*) is a novel regulator of regenerative capacity

Having established a robust assay to examine the effect of manipulating gene activity on regeneration following ablation, we explored the potential for identifying novel regulators of regeneration. We chose to investigate the uncharacterized gene *CG9572* because we have experimentally validated a DRMS enhancer in its vicinity and because its expression was one of the most strongly upregulated by damage in regenerating cells according to the ***Khan et al., 2017*** data set. *CG9572* is predicted to encode a 441 amino acid protein of unknown function. In order to better characterize CG9572, we performed protein blast (blastp) and alignment scoring, which revealed strong sequence similarity to the EGF domain repeats of the Jagged protein (***Figure 5—figure supplement 1A***), a membrane-bound ligand for Notch in vertebrates (***Lindsell et al., 1995***). A Jagged ortholog in *Drosophila* has not been described. The peptide sequence is predicted to contain seven EGF-type repeats that each displays a characteristic spacing of cysteine residues (***Figure 5—figure supplement 1B–C***). EGF-like repeats are found in all Notch ligands that have been described to date (reviewed by ***Kovall et al., 2017***). Similar to Jagged, CG9572 also has a 14-amino acid hydrophobic stretch close to its N-terminus that is likely to function as a signal peptide (***Figure 5—figure supplement 1D***). However, unlike Jagged or other Notch ligands it lacks a second hydrophobic stretch that would serve as a transmembrane domain, or a Delta/Serrate/LAG-2 (DSL) domain characteristic of Notch ligands (***Figure 5—figure supplement 1B***). Thus, the predicted CG9572 protein has sequence similarity to Jagged, but unlike Jagged or other known Notch ligands, it is likely to be secreted. Due its similarity to mammalian Jagged, we have called this protein Asperous (Aspr). When a tagged version, *UAS-aspr::HA*, was expressed in the posterior compartment using *en-GAL4*, HA staining appeared to be cytoplasmic (***Figure 5B***), localizing towards the apical surface of the disc proper (***Figure 5C***) and with punctae observed in the anterior compartment. When the stronger *hh-GAL4* driver was used, punctae were observed throughout the anterior compartment consistent with the Aspr protein being secreted from cells (***Figure 5D***). Overexpression of *aspr* in the whole wing

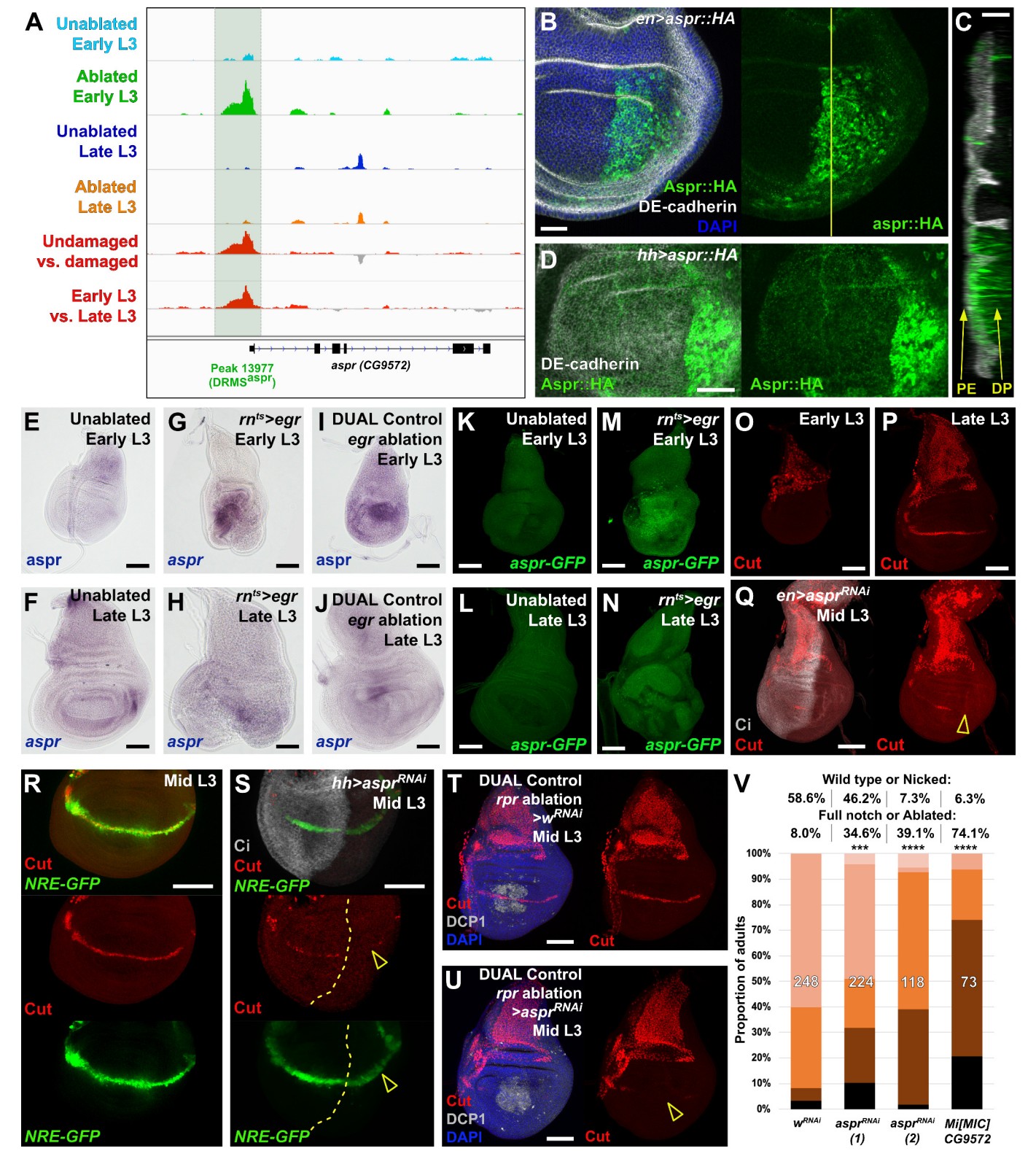

**Figure 5.** Uncharacterized gene *CG9572/asperous* is a novel regulator of wing disc regeneration. (**A**) ATAC-seq chromatin accessibility traces at the *asperous* (*aspr*) locus, showing the computationally detected DRMS peak that was experimentally validated as a DRMS enhancer (DRMS[aspr], dark green box), (**B**) Ectopic expression of an epitope tagged Aspr (Aspr::HA, green) in the posterior compartment using *en-GAL4*. The protein is cytoplasmic and

*Figure 5 continued on next page*

Figure 5 continued

is mostly excluded from nuclei. DE-cadherin (gray). DAPI: blue. Yellow line indicates plane of image in (C), (C) Z-section though the disc shown in (B), showing mostly apical localization of Aspr::HA in the expressing cells (green), and presence of Aspr::HA between the disc proper (DP) and peripodial epithelium (PE), suggesting extracellular localization. DE cadherin shows cell membranes (gray), (D) High magnification imaging of an apical disc section of a disc expressing Aspr::HA (green) under the control of *hh-GAL4*, showing punctae of staining away from the expressing cells at a level between the peripodial membrane and the disc proper epithelium, (E–F) RNA *in situ* hybridization detects weak *aspr* expression in the ventral and lateral areas of the disc in early L3 (E) and late L3 (F) discs, but is mostly absent from the pouch, (G–J) RNA *in situ* hybridization detecting *aspr* in early L3 (G and I) or late L3 (H and J) discs following ablation with *egr* using *rn^{ts}>* (G–H) or DUAL Control (I–J). In both cases, *aspr* is upregulated in blastema cells in the pouch upon ablation in early L3 discs, but only weakly in late L3, (K–N) Expression of an *aspr* GFP MiMIC reporter line (green) in early L3 (K and M) and late L3 discs (L and N) during normal development (K–L) and following ablation with *rn^{ts}>egr* (M–N). Consistent with the RNA *in situ* expression data, *aspr* reporter activity is mostly absent during normal development, being activated by damage in early L3 discs, with reduced activity in late L3 discs, (O–P) Expression of *cut* during normal development in early L3 (O) and late L3 (P) wing discs. Cut protein (red) is detected in cells of the notum, including myoblasts and trachea, at both developmental time points, and becomes upregulated at the D-V boundary in late L3 in response to Notch signaling, (Q) Knockdown of *aspr* in the posterior compartment with *en-GAL4* driving *aspr* RNAi delays onset of *cut* expression, as shown by a lack of Cut at mid L3 (open arrowhead) when it usually extends across the entire posterior compartment. Ci: Gray, (R–S) Cut protein (red) and a Notch reporter *NRE-GFP* (green) in mid-L3 wing discs, showing expression of both extending across the entire D-V boundary in wild type discs (R), while discs expressing *aspr* RNAi in the posterior compartment using *hh-GAL4* have delayed expression of Cut and weaker *NRE-GFP* activity (S, open arrowheads). Ci (gray) demarcates A-P compartment boundary (yellow dotted line), (T–U) Mid L3 discs ablated with *rpr* using DUAL Control *DVE>>GAL4* and driving a control RNAi (T) or *aspr* RNAi (U). Cut protein (red) is quickly reestablished following *rpr* ablation with a control RNAi, and extends across the D-V boundary (T). Knockdown of *aspr* in the entire pouch with *DVE>>GAL4* prevents reestablishment of Cut during regeneration (U, open arrowhead). DCP1: gray, DAPI: blue, (V) Knockdown of *aspr* using two different RNAi lines, or heterozygosity for a presumed *aspr* mutant (*Mi[MIC]CG9572*), in discs ablated with *rpr* using DUAL Control *DVE>>GAL4* reduces regeneration compared to $w^{RNAi}$ control, as assessed by wing size. *** p = 0.0007 ($aspr^{RNAi}$ (1)), **** p <0.0001, Fisher exact test. Scale bars in (B–D) = 20 μm, other scale bars = 50 μm.

The online version of this article includes the following figure supplement(s) for figure 5:

**Figure supplement 1.** *asperous* (*aspr*, *CG9572*) peptide sequence is consistent with a non-membrane bound Notch ligand.

**Figure supplement 2.** *aspr* promotes regeneration.

pouch using *rn-GAL4* does not result in an observable phenotype in the adult wing (*Figure 5—figure supplement 2A*), but expression in the posterior compartment infrequently causes abnormal folding in the pouch at the boundary with wild type cells (*Figure 5—figure supplement 2C*). However, this did not result in an adult wing phenotype (data not shown).

As the annotated transcriptional start site of *aspr* is close to a DRMS peak (*Figure 5A*), we examined its expression following damage in discs of different maturity using RNA *in situ* hybridization, and a gene-trap cassette insertion line (*Mi[MIC]CG9572^{[MI02471]}*), which bears an eGFP gene that can be used to monitor *aspr* expression (*Venken et al., 2011*). In undamaged L3 wing discs, *in situ* hybridization shows *aspr* is expressed at low levels in the ventral and lateral areas of the disc at low levels (*Figure 5E–F*). Similarly, the gene trap shows little to no expression (*Figure 5K–L*). Upon damage in early L3 discs, *aspr* is upregulated strongly in the region of the blastema, as shown in discs ablated by both *rn^{ts}>egr* (*Figure 5G,M*) and DUAL Control ablation with *egr* (*Figure 5I*). In damaged discs from late L3 larvae, *aspr* has much weaker damage-induced expression (*Figure 5H,J,N*). Knockdown of *aspr* with two different RNAi lines in the developing wing pouch using *rn-GAL4* in undamaged discs has little effect on adult wing size or patterning (*Figure 5—figure supplement 2B* and data not shown). However, we found that knockdown of *aspr* in mid L3 discs using *en-GAL4* delays the onset of expression of the Notch target *cut* at the prospective wing margin in the posterior compartment (*Figure 5O–Q*), suggesting *aspr* might promote Notch signaling during normal development. This is also shown by the weak reduction in fluorescence of a Notch reporter, *NRE-GFP* (*Zacharioudaki and Bray, 2014*; *Figure 5R–S*). These data suggest that Aspr is a secreted regulator of Notch signaling in the wing, which is strongly activated in regenerating tissue upon damage. To address whether *aspr* is necessary for regeneration, we used DUAL Control to reduce its expression in regenerating cells in mid-L3 discs using the two different RNAi lines following *rpr* ablation of the wing pouch. At this stage Cut expression at the margin is beginning to be established (*Figure 5R*), while ablation at an earlier stage leads to a delay in development and hinders Cut expression. Upon *aspr* knockdown the presence of Cut in these discs is markedly reduced (*Figure 5T–U*), suggesting that ablation combined with the loss of *aspr* may limit Cut expression that has already been initiated. The extent of regeneration is also decreased (*Figure 5V*). This effect on regeneration was also observed with *rn^{ts}>egr* ablation (*Figure 5—figure supplement 2D*). These

experiments suggest that *aspr* promotes specification of the wing margin during development. We also tested the MiMIC line that we used as a GFP reporter for *aspr* expression, which is likely to also be an *aspr* mutant due to the mutagenic cassette in the insertion that is designed to disrupt gene expression (*Venken et al., 2011*). The insertion is in the first intron of the coding region, downstream of the transcriptional start site of all three *aspr* transcripts (*Figure 5—figure supplement 2E*). Semi-quantititive PCR of *rn^ts^>egr* ablated discs from MiMIC hemizygous animals (*Mi[MIC] CG9572^[MI02471]^/Y*) showed strongly reduced levels of *aspr* mRNA compared to ablated wild type discs (*Figure 5—figure supplement 2F*), consistent with this line being a transcriptional mutant. *aspr* hemizygous animals have no obvious developmental defects but show a strongly decreased ability to regenerate when ablated with DUAL Control (*Figure 5V*) and *rn^ts^>egr* (*Figure 5—figure supplement 2G*). Expression of Aspr::HA does not strongly affect regeneration (*Figure 5—figure supplement 2H*), indicating that while it is necessary for regeneration, ectopic expression alone it is not sufficient to improve it, as we have demonstrated with other pro-regeneration genes such as *wg* (*Harris et al., 2016*). Thus, we have used proximity to a DRMS enhancer to identify a novel gene with sequence similarity to Notch ligands as a potential regulator of regenerative capacity.

## Manipulating regulators of JAK/STAT signaling and chromatin accessibility restores regenerative capacity in mature discs

From our ATAC-seq dataset we noted that the strongest damage-responsive change in chromatin accessibility of all detected DRMS peaks was at the *apontic* (*apt*) gene (*Figure 2D*, *Figure 6A* and *Supplementary file 3*). We have shown that DNA spanning this peak (Peak 4931) does indeed function as a DRMS enhancer *in vivo* (*Figure 2G–G'*), while the data of Khan et al. shows that *apt* is transcriptionally upregulated in blastema cells (*Khan et al., 2017*). *apt* is also known as *trachea defective* (*tdf*), and encodes a b-Zip transcription factor that has been characterized in tracheal development (*Eulenberg and Schuh, 1997*; *Gellon et al., 1997*) and formation of both male and female germlines (*Starz-Gaiano et al., 2008*; *Monahan and Starz-Gaiano, 2016*). In the male germline, decreasing *apt* function results in increased expression of JAK/STAT targets, suggesting *apt* functions, at least in this context, to restrict JAK/STAT signaling (*Starz-Gaiano et al., 2008*; *Monahan and Starz-Gaiano, 2016*), which is known to be required for wing disc regeneration (*Katsuyama et al., 2015*; *Figure 4—figure supplement 1E*). More recently, *apt* has been shown to promote expression of *cyclin E* and *hedgehog* in imaginal discs (*Liu et al., 2014*; *Wang et al., 2017*). Using an antibody that recognizes the Apt protein (*Liu et al., 2003*), we found that Apt is expressed at high levels in the squamous cells of the peripodial epithelium and tracheal tubes of undamaged developing wing discs throughout L3, and expression is not obviously above background in cells of the disc proper (*Figure 6—figure supplement 1A–B*). However, optical section imaging of ablated discs shows that Apt can be detected in the disc proper following damage in early L3 (*Figure 6B*), but not in late L3 (*Figure 6C*). Thus, the presence of a DRMS peak within the *apt* gene is consistent with its stage-specific damage-responsive expression.

To test whether *apt* is necessary for regeneration, we reduced its expression using DUAL Control by expressing an *apt* RNAi transgene in regeneration-competent early L3 discs. In view of the demonstration that *apt* can promote wing growth by promoting Hedgehog expression (*Wang et al., 2017*), we were surprised to find that the adult wings were more complete than in ablated controls not expressing the *apt* RNAi (*Figure 6D*). We then examined the effect of *apt* knockdown in late L3 discs and found that, once again, adult wings were more complete than in controls (*Figure 6D*). These observations indicate that *apt* knockdown either protects discs from damage during the ablation phase or promotes regeneration. Since the levels of cleaved caspase DCP1 appeared similar in the presence and absence of *apt* knockdown (*Figure 6E–F*), this suggests that *apt* normally acts to limit regeneration following damage. This is corroborated by a hypomorphic *apt* mutant (*apt^K15608^/+*, [*Eulenberg and Schuh, 1997*]), which also shows a mild increase in regenerative ability compared to wild type at both time points when ablated with DUAL Control and *rn^ts^>egr* (*Figure 6—figure supplement 1C–D*). As *apt* is known to act in the germline to limit JAK/STAT signaling, we examined whether it might affect regeneration by influencing JAK/STAT activity. We observed the expression of a fluorescent Stat92E reporter (*STAT-GFP*) in late L3 discs ablated with DUAL Control in the presence of *apt* knockdown. Normally at this stage, STAT activity is minimal in regenerating cells of ablated discs (*Figure 6E*). However, with *apt* knockdown, we found an increase in the activity of this reporter specifically in blastema cells (*Figure 6F*). This is more clearly visible

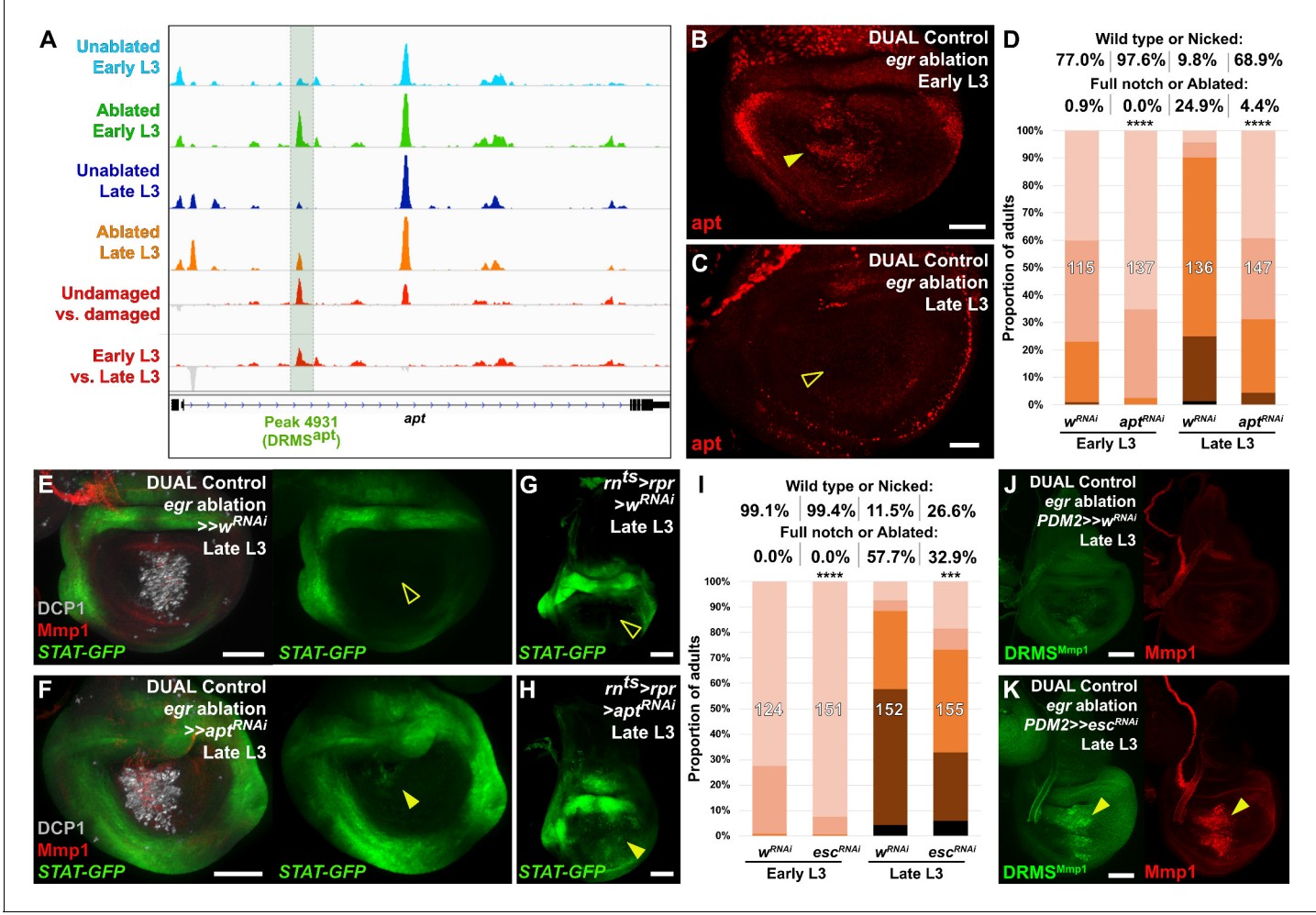

**Figure 6.** The DRMS enhancer-regulated gene *apontic* (*apt*) and the chromatin regulator *esc* can be manipulated to augment regenerative capacity in mature discs. (**A**) ATAC-seq chromatin accessibility traces at the *apt* locus, showing the computationally detected DRMS peak that was experimentally validated as a DRMS enhancer (DRMS^apt, dark green box), (**B-C**) Apt protein (red) detected in the disc proper in early L3 (**B**) and late L3 (**C**) discs ablated with *egr* using DUAL Control. *Apt* is expressed in blastema cells upon ablation in early L3 discs (arrowhead) but is absent in late L3 (open arrowhead). Developmental expression of *apt* persists in cells of the peripodial epithelium and trachea at both time points, (**D**) RNAi knockdown of damage-induced *apt* with *DVE>>GAL4* in discs ablated with *rpr* using DUAL Control increases regeneration in both early and late L3 discs compared to w^RNAi control, as assessed by wing size. **** p <0.0001, Fisher exact test, (**E-H**) Late L3 discs bearing a *Stat92E* reporter (*STAT-GFP*) ablated with DUAL Control *DVE>>* using *egr* (**E-F**) or *rn^ts>rpr* (**G-H**) in the presence of control RNAi (**E and G**) or *apt* RNAi (**F and H**). Ablation in control samples shows no increase in reporter activity in the regenerating cells of late L3 discs (**E and G**, open arrowheads), while knockdown of *apt* in the pouch results in increased Stat92E reporter expression in cells surrounding the zone in the medial disc where ablation occurs using DUAL Control (**F**, arrowhead), or in the majority of the pouch when ablated with *rn^ts>rpr*, which targets the entire pouch (**H**, arrowhead). GFP expression from earlier developmental expression persists in the hinge region of all samples. DCP1: gray, Mmp1:red, (**I**) RNAi knockdown of damage-induced *esc* with *PDM2>>GAL4* in discs ablated with *rpr* using DUAL Control increases regeneration in both early and late L3 discs compared to w^RNAi control, as assessed by wing size. **** p <0.0001, Fisher exact test. (**J-K**) Late L3 discs bearing DRMS^Mmp1 GFP reporter (green) ablated with DUAL Control using *egr*. Control discs expressing a control RNAi (**J**) show little damage-induced reporter activity or *Mmp1* expression (red). By comparison, knockdown of *esc* by *PDM2>>GAL4* (**K**) increases damage-induced reporter expression and *Mmp1* (arrowheads). Scale bars in (**B**) and (**C**) = 20 µm, other scale bars = 50 µm. The online version of this article includes the following figure supplement(s) for figure 6:

**Figure supplement 1.** *apt* mutant discs have increased regenerative capacity.

**Figure supplement 2.** The chromatin regulator *extra sex combs* (*esc*) limits regenerative capacity in part via DRMS silencing.

with *rn^ts>egr* ablation, which damages a larger area of the disc (*Figure 6G–H*). These data suggest that rather than being necessary for regeneration, activation of *apt* expression in young, but not old discs, might be a mechanism of tempering the extent of regeneration when it occurs.

The ability to augment regeneration by manipulating a single gene is striking, but could be explained if this manipulation has pleiotropic effects on key aspects of the regeneration process. Indeed, JAK/STAT signaling, which our data suggest can be regulated by *apt* in this context, has been described as a central regulator of regeneration via its influence on both local blastema formation and systemic physiological responses (*Katsuyama et al., 2015*). As DRMS enhancers are associated with multiple genes, we hypothesized that manipulating levels of chromatin silencing factors that are involved in inactivating these enhancers could restore the damage-responsive expression of their target genes in mature discs, increasing regenerative capacity. To this end, we used DUAL Control to knock down a panel of epigenetic silencing factors in regenerating tissue of late L3 discs. Most of those tested did not have an obvious effect (*Figure 6—figure supplement 2A*). However, inhibition of the Polycomb group gene *extra sex combs* (*esc*), which encodes a component of the PRC2 complex (reviewed by *Kassis et al., 2017*), specifically during the regeneration period consistently improved regeneration, even in discs ablated in early L3 (*Figure 6I*). We used the weaker *PDM2*-driven version of DUAL Control for these experiments to avoid the negative effects associated with strong or widespread knockdown of these epigenetic regulators. Under these conditions, we observed no adult wing defects when *esc* was knocked down in unablated wings for this short duration (data not shown). Thus, even a small reduction in *esc* levels is likely to improve regeneration. Ablation in a heterozygous *esc* background ($esc^5/+$) similarly leads to improved regeneration when ablated with $rn^{ts}>egr$ (*Figure 6—figure supplement 2C*), although this effect was not significant when ablated using DUAL Control, unlike targeted knockdown with RNAi (*Figure 6—figure supplement 2B*).

To test whether the improved regenerative capacity of *esc* knockdown was due to alteration of DRMS activity, we examined the activity of three different DRMS reporters, DRMS$^{Wnt}$, DRMS$^{Mmp1}$, and DRMS$^{aspr}$, in late L3 discs expressing *esc*$^{RNAi}$. The level of GFP expression from all of the reporters was increased in ablated discs with *esc* knockdown compared to controls, as was *wg* and *Mmp1* expression (*Figure 6J–K* and *Figure 6—figure supplement 2D–G*), indicating that the loss of *esc* leads to improved regeneration in part via reactivation of multiple DRMS enhancers. Thus, reducing Polycomb-mediated silencing appears capable of overcoming the suppression of a regeneration program that operates simultaneously at multiple loci in the genome.

## Discussion

We previously showed that the reduced expression of *wg* and *Wnt6* elicited by damage as discs mature can be explained by the properties of a damage-responsive and maturity-silenced (DRMS) enhancer that lies between the two genes (*Harris et al., 2016*). Manipulating this enhancer by deleting the maturity silencing module restores damage-responsive *wg* expression in mature discs, but this alone is insufficient to preserve regenerative capacity at this stage of development. These observations suggest that genes other than *wg* are required for regeneration, and might be regulated by in a similar way. Now, we have identified three additional genes, *Mmp1*, *apt* and *aspr*, which function in regeneration and are regulated by DRMS enhancers. The damage-responsive expression of each of these genes is reduced in late L3 discs compared to early L3 discs, while DUAL-Control ablation experiments showed that the function of two of these genes, *Mmp1* and *aspr*, in the blastema contributes to regeneration. Additionally, a negative regulator of regeneration, *apt*, seems to be regulated in a similar way. Moreover, our ATAC-seq data reveal a number of other regions within the genome with chromatin accessibility signatures that are suggestive of being DRMS enhancers that could potentially regulate multiple other genes that function during regeneration. Since the manipulation of a chromatin silencing factor, *esc*, can de-repress at least three DRMS enhancers, and also promote regeneration, it is likely that the overall effect of the expression of multiple genes regulated by DRMS enhancers is to promote regeneration. However, it is important to note that the number of regions we identified using ATAC-seq are likely to be an underestimate since our assay examined chromatin changes using whole discs, in which only a small proportion of cells are undergoing regeneration. In the future it will be important to test whether additional regions of interest, and potential target genes, might be uncovered through the use of single-cell ATAC-seq or similar approaches. This information will also be key to understanding the degree to which DRMS enhancers contribute to the overall regulation of regeneration in imaginal discs.

As our approach likely identified only the regions that changed most significantly in response to damage, it is striking that we identified an uncharacterized gene (*CG9572/aspr*) and a gene previously not described in regeneration (*apt*) as among the top hits. Knockdown of *aspr* adversely impacted regenerative growth but produced no obvious effect on the normal development of the wing except perhaps a delay in the development of the wing margin. We have shown that following *aspr* knockdown, expression of the markers at the margin such as Cut and the Notch reporter was delayed both in normal and regenerating discs, suggesting that *aspr* could potentially promote Notch signaling. Our analysis reveals that Aspr has sequence similarity to the vertebrate Notch ligand Jagged (*Lindsell et al., 1995*), raising the possibility that its influence on Notch signaling could be as a ligand. However, unlike most Notch ligands characterized to date, Aspr lacks a DSL domain or a predicted transmembrane domain. Non-canonical Notch ligands that lack a DSL domain have been identified in mammals (*Baladrón et al., 2005*; *Falix et al., 2012*) and have been shown to regulate Notch signaling when engineered into *Drosophila* (*Bray et al., 2008*), while several Notch ligands have been characterized in *C. elegans* that lack transmembrane domains (*Chen and Greenwald, 2004*; *Komatsu et al., 2008*). The lack of a transmembrane domain suggests that Aspr is likely secreted. Consistent with this notion, we observed punctate staining outside the region of expression when *aspr::HA* was overexpressed. Being anchored to the membrane is thought to be crucial for the mechanism by which Notch ligands function in *Drosophila*; endocytosis of the ligand engaged to Notch is presumed to generate the mechanical force that alters Notch conformation and renders it susceptible to proteolytic cleavage, which eventually results in nuclear translocation of the intracellular domain (*Parks et al., 2000*; *Langridge and Struhl, 2017*). Thus, at least based on this view of Notch signaling, it is unlikely that Aspr could function as a canonical Notch ligand. Moreover, Notch signaling is thought to function as part of double repressive mechanism (*Herranz et al., 2008*) that controls regenerative growth (*Smith-Bolton et al., 2009*), and is reduced in blastema cells to permit *Myc*-dependent growth. As such, it is unclear how Aspr might promote Notch signaling, and yet also be required for regeneration. Importantly, overexpression of *aspr* did not appear to increase Notch signaling, at least as assessed by activation of the Notch reporter. Therefore, the role of Aspr in promoting regeneration could be independent of a direct influence on Notch signaling and remains to be explored further.

In addition to *aspr*, we also found that the transcriptional regulator *apt* functions in regeneration, although in contrast to *aspr*, knockdown of *apt* appears to improve regeneration in discs ablated in either early or late L3. Knockdown of *apt* increased expression of a STAT reporter, suggesting this improvement might be via increased JAK/STAT signaling. However, we cannot exclude the possibility that other mechanisms might be more important. A previous study has shown that reducing JAK/STAT signaling reduced tissue loss following *egr*-induced ablation, but did not seem to compromise the extent of compensatory proliferation (*La Fortezza et al., 2016*). Furthermore, the overexpression of *upd1* or *upd2* concurrently with ablation did not improve regeneration. However, increasing Stat92E activity in the blastema during the regeneration phase was not tested in that study, as appears to occur when *apt* is knocked down using DUAL Control.

We also found that reducing PcG-mediated repression via knockdown of *esc* improved overall regeneration. Reducing *esc* activity derepressed at least three DRMS enhancers that were tested using reporter gene constructs. Since a reduction in *esc* levels alleviates silencing at multiple DRMS enhancers, it is likely that many of these other enhancers are also regulated by Polycomb-mediated silencing, as for DRMS$^{Wnt}$. These findings parallel those of other groups who have used chromatin profiling in imaginal discs to show that enhancer accessibility is an important mechanism that regulates various properties of imaginal discs, including specification of their identity during development and cellular proliferation (*McKay and Lieb, 2013*; *Ma et al., 2019*). In a recent study that investigated the basis of terminal exit from the cell cycle in cells of the *Drosophila* wing disc (*Ma et al., 2019*) it was found that distal enhancers for key cell cycle regulators such as *cyclin E* and *string* become less accessible as development proceeds. This occurs independently of cell-cycle status but instead seems to be governed by a temporal program. The robust exit from the cell-cycle could potentially be explained by reduced levels of many cell-cycle regulators caused by the silencing of enhancers at multiple loci. Similarly, an enhancer-mediated mechanism that reduces the expression of multiple genes necessary for regeneration in mature discs might be more robust than one that relies on reduced expression of a single key regulator.

If regeneration in maturing discs is indeed suppressed by the silencing of multiple DRMS enhancers that regulate genes promoting regeneration, it could be restored, at least in principle, in two different ways. One would be by alleviating the silencing at multiple DRMS enhancers; knockdown of *esc* likely functions in this way. Another way would be by activating pathways that function downstream of genes that promote regeneration (bypass suppression). Our previous work (*Worley et al., 2018*) has shown that the JAK/STAT pathway can promote plasticity during regeneration. Knockdown of *apt* increases expression of the STAT reporter. Perhaps this still poorly-understood mechanism acts in parallel to the alleviation of PcG-mediated silencing.

Damage-responsive enhancers have been identified in multiple species and found to share common regulators, including JNK signaling. The *lepb* gene in zebrafish appears to be regulated by such an enhancer, which is altered epigenetically following injury to fins or the heart, and requires AP-1 binding sites for activation of gene expression (*Kang et al., 2016*). In mice, discrete regulatory elements that regulate muscle regeneration, bone fracture and soft-tissue injury have also been described (*Guenther et al., 2015*; *Aguilar et al., 2016*). However, it is not known whether these enhancers become less active with maturity since zebrafish preserve their regenerative capacity and adult mice continue to heal muscle and fractured bones. In contrast, the murine neonatal heart is known to lose regenerative ability within the first week of life (*Porrello et al., 2011*). Thus, it will be interesting to see whether genes implicated in cardiac regeneration in neonatal mice are regulated by similar enhancers that become silenced with maturity. Finally, there is evidence that damage-responsive regulatory elements are conserved in the human genome (*Suzuki et al., 2019*) and function in models of human epithelial injury (*Lander et al., 2017*). Thus, the study of these enhancers and the mechanism by which they regulate the response to injury could provide important insights in human regenerative medicine in future.

## Materials and methods

### Fly stocks and genotypes

Stocks and crosses were maintained on yeast food at 25°C, except those for GAL4/UAS based ablation experiments, which were maintained at 18°C. Stocks used in this study: *rn^ts^>rpr/egr* (*w1118;; rn-GAL4, tub-GAL80ts, UAS-rpr* or *UAS-egr*), *rn^ts^>* (*w1118;; rn-GAL4, tub-GAL80ts*)(*Smith-Bolton et al., 2009*), *AP-1-RFP* (*Chatterjee and Bohmann, 2012*), *UAS-his::RFP* (*Emery et al., 2005*), *UAS-dGFP* and *lexAOp-dGFP* (*Lieber et al., 2011*), *UAS-dILP8* (*Colombani et al., 2012*), *DRMS^Wnt^-GFP* (*BRV118-GFP*, [*Harris et al., 2016*]), *PCNA-GFP* (*Thacker et al., 2003*), *hh-GAL4* (*Tanimoto et al., 2000*), *NRE-GFP* (*Zacharioudaki and Bray, 2014*), *UAS-aspr^SIRNAi(M1)^* (labeled as (2) in manuscript) and *UAS-aspr::HA* (generous gifts from David Bilder). Stocks obtained from the Bloomington stock center: *UAS-hep^CA^* (BL6406), *hep^r75^* (BL6761), *pc^15^* (BL24468), *lexAOp-hrGFP* (BL29954), *10xStat92E-GFP* (BL26197) *UAS-w^RNAi^* (BL33613), *UAS-myc* (BL9674), *UAS-myc^RNAi^* (BL51454), *UAS-upd1^RNAi^* (BL33680), *UAS-upd2^RNAi^* (BL33949), *UAS-upd3^RNAi^* (BL32859), *UAS-dome^RNAi^* (BL53890), *UAS-stat92E^RNAi^* (BL33637), *UAS-apt^RNAi^* (BL26236), *UAS-wg^RNAi^* (BL32994), *UAS-mmp1^RNAi^* (BL31489), *UAS-su(z)2^RNAi^* (BL57466), *UAS-e(pc)^RNAi^* (BL67921), *UAS-Sfmbt^RNAi^* (BL32473), *UAS-ph-p^RNAi^* (BL33669), *UAS-pc^RNAi^* (BL33622), *UAS-psc^RNAi^* (BL38261), *UAS-pho^RNAi^* (BL42926), *UAS-su(z)12^RNAi^* (BL33402), *UAS-esc^RNAi^* (BL31618) *UAS-Sp1^RNAi^* (BL35777), *UAS-CG9572/aspr^RNAi^* (labeled as (1) in manuscript) (BL58340), *en-GAL4* (BL30564), *Mi[MIC]CG9572 [MI02471]* (BL 35863), *apt^K15608^* (BL10455), *esc^5^* (BL3142).

### Ablation experiments

GAL4/UAS-based genetic ablation experiments and wing scoring were performed essentially as described in *Smith-Bolton et al., 2009*, with each experimental condition compared to a suitable control that was ablated and scored in parallel. Unless otherwise indicated, discs were dissected and fixed for immunofluorescence immediately after the ablation period. DUAL Control ablation experiments were also density controlled (50 larvae per vial) and experiments conducted at 25°C, with a 37°C heat shock administered at day 3.5 or day 4.5 in a circulating water bath for 45 min unless otherwise stated. A detailed ablation protocol is available upon request. Discs were dissected, fixed and stained at 24 hr PHS unless otherwise stated, with at least 50 discs per genotype or condition. Wing scoring experiments were performed on the number of flies per genotype shown. Statistical

significance was evaluated using Fisher exact tests on raw numbers of offspring in each class of regeneration totaled from each experiment compared to its accompanying control. Stacked bar charts of each regeneration scoring experiment are presented as separate bar graphs with error bars in *Supplementary file 5*. Physical wounding experiments were performed essentially as described for *ex vivo* culture in *Harris et al., 2016*.

## DUAL Control stock generation

The DUAL Control stock genotype is *hsFLP; hs-p65::zip, lexAOp-ablation/CyO; salm-zip::LexA-DBD, PDM2 or DVE>>GAL4*. A non-ablating stock was generated without a *lexAOp-ablation* transgene. The ablation drivers used were *lexAOp-rpr* and *lexAOp-egr*. Each was generated by replacing the *GFP* coding sequence from *pJFRC19-13XLexAop2-IVS-myr::GFP* (*Pfeiffer et al., 2010*) with the full *rpr* or *egr* coding sequence from genomic DNA and LP03784 respectively. The resulting transgenes were inserted into landing site *su(Hw)attP5* (BL32231) via PhiC31 recombination. The *hs-p65* construct was built by cloning nucleotides −242 to 0 upstream of the TSS from the *Hsp70* gene into *pAttB* along with the *p65AD::zip* and *Hsp70* 3′UTR from *pBPp65ADZpUw* (*Pfeiffer et al., 2010*). The transgene was inserted into landing site attP40 (BL25709) and recombined with lexA*Op-rpr* or *lexAOp-egr*. The *LexA-DBD* was generated by removing the *GAL4-DBD* sequence from *pActPL-zip:: GAL4-DBD* (*Luan et al., 2006*), and replacing it with the codon optimized *LexA-DBD* from *pBPLexA::p65Uw* (*Pfeiffer et al., 2010*). This *zip::LexA-DBD* cassette was then cloned into *pAttB* along with the *salm* enhancer fragment R85E08 (Flylight), the DSCP sequence (*Pfeiffer et al., 2008*) and *Hsp70* 3′UTR. This transgene was inserted into landing site *attP2* (BL8622). The *PDM2>>GAL4* and *DVE>>GAL4* constructs were generated by cloning the *PDM2* or *DVE* enhancer fragments R11F02 or R42A07 (Flylight) into *pAttB*, along with an *FRT-PolyA-FRT* cassette, the *GAL4* coding sequence (GenBank: NM_001184062) and *SV40* 3′UTR. The transgene was inserted into landing site VK00027 (BL9744) and recombined with *salm-zip::LexA-DBD*. Both recombined chromosomes were built into a single stock with hsFLP (BL8862) on the X chromosome. Detailed plasmid maps are available on request.

## Transgenic reporter line construction

The *Mmp1-GFP* enhancer reporter was generated by amplifying the *Mmp1* genomic region using primers listed in *Supplementary file 4* and cloning upstream of a minimal *hsp70* promoter and *eGFP* coding sequence into *pAttB* (accession KC896839.1). Related *Mmp1* GFP reporters were generated by replacing the *Mmp1* enhancer DNA with genomic regions amplified from genomic DNA with the primers listed in *Supplementary file 4*, as were reporters for the DRMS[ftz-f1], DRMS[bru2], DRMS[aspr/CG9572], and DRMS[apt] regions. All GFP reporters were inserted into the *AttP40* landing site via PhiC31 recombination ensuring comparability. Transgenic services were provided by BestGene (Chino Hills, CA).

## Immunofluorescence and *in situ* hybridization

Discs were fixed and stained for immunofluorescence essentially as in *Harris et al., 2016*, with at least 50 discs per genotype or condition. Samples were mounted in ProLong Gold Antifade Reagent (Cell Signaling, Beverly, MA). The following primary antibodies were used in this study: from the DSHB, Iowa City, IA; mouse anti-Wg (1:100, 4D4), mouse anti-Mmp1 (1:100, a combination of 14A3D2, 3A6B4 and 5H7B11), rat anti-Ci (1:10, 2A1). anti-Cut (1:100, 2B10). Other antibodies; rabbit anti-DCP1 (1:1000, Cell Signaling), rabbit anti-TDF/apt (1:1000, *Liu et al., 2003*), rabbit anti-HA (1:1000, Cell Signaling). Secondary antibodies used were from Cell Signaling, all at 1:500; donkey anti-mouse 555, donkey anti-rabbit 555, donkey anti-rat 647, donkey anti-rabbit 647, donkey anti-rabbit 488 and donkey anti-mouse 488. Nuclear staining was by DAPI (1:1000, Cell Signaling). Samples were imaged on a Leica TCS SP5 Scanning Confocal, Zeiss LSM 700 Scanning Confocal or Zeiss M2 Imager with Apotome. RNA *in situ* hybridizations were performed according to established methods for alkaline phosphatase-based dig-labeled probe detection. Discs were dissected and fixed as for immunofluorescence, Digoxigenin labeled probes were generated targeting the *aspr* gene coding sequence using the primer pairs listed in *Supplementary file 4* to generate templates with T7 sequences at either the 5′ (sense probe) or 3′ (anti-sense probe) ends. Control and

experimental discs were stained simultaneously for the same duration, mounted in Permount (Fisher Scientific, Pittsburg, PA) and imaged on a Zeiss Axio Imager M2.

## Semiquantitative PCR

Around 50 discs were dissected from equivalently staged male larvae of genotypes $rn^{ts}$>egr (ablated), $rn^{ts}$> (unablated), rn-GAL4/UAS-aspr:HA or Mi(MIC)CG9572/Y; ; $rn^{ts}$>egr (ablated) at early L3 (day 7, 18°C), and added to equal volumes of Trizol (Sigma). RNA was extracted according to established protocols, yielding 2–4 µg total RNA per sample. cDNA was synthesized using Denville rAmp cDNA Synthesis Kit (Thomas Scientific) from 200 ng starting RNA of each sample with a polyT primer. Subsequently, 1 ul of each cDNA library was used for reduced-cycle semi-quantitative PCR with the primers listed in *Supplementary file 4* to detect *aspr* and *actin* expression. The entire assay was repeated using discs from a separate ablation experiment for confirmation.

## ATAC-seq and sequencing analysis

Samples for ATAC-seq library preparation were generated as follows: Larvae of genotype +; +; rn-GAL4, GAL80ts ($rn^{ts}$>, undamaged) and +; +; rn-GAL4, GAL80ts, UAS-egr ($rn^{ts}$>egr, damaged) were grown to early L3 (day 7) or late L3 (day 9) and upshifted to 30°C for 40 hr, as for other ablation experiments. Discs were dissected in PBS immediately upon downshift and collected as pools of 100 discs for early L3 samples and 50 discs for late L3 samples. The four samples were placed in lysis buffer (10 mM Tris 7.5, 10 mM NaCl, 3 mM MgCl2, 0.1% IGEPAL CA-630), pelleted, and exposed to the Tn5 transposase enzyme (Illumina) essentially as in *Buenrostro et al., 2013*. Three biological repeats were performed, and DNA was sequenced on a HiSeq2500 or HiSeq4000 as single index, multiplexed samples with 50PE or 100PE reads. Reads were trimmed to 50 bp using cutadapt (*Martin, 2011*) and then aligned to the dm3 reference genome using bowtie 2 (setting: `-seed` 123, -q –X 2000). Reads with quality scores below five were removed. Reads mapping to Chr2L/2R/3L/3R/4/X were employed in subsequent analysis. Peaks were called with MACS2 (setting: `-g dm -keep-dup all -shift 9 -nomodel -seed` 123), using a sonicated genomic DNA dataset as a control (*Zhang et al., 2008*). Signal tracks were generated from individual replicates using Deeptools v2.4.1 (*Ramírez et al., 2016*). Signal in browser shots are represented as z scores of pooled replicates as described previously (*Uyehara et al., 2017*).

## Differential accessibility analysis

ATAC peak calls from pooled datasets were ranked by MACS2 q-score. The top 11,500 peaks from each dataset were selected and combined into a union peak set of 46,000 peaks which was subsequently reduced by merging any peaks that overlapped by 1 bp or more using the GenomicRanges Bioconductor package (*Gentleman et al., 2004*; *Lawrence et al., 2013*). This resulted in a final union peak set of 14,142 peaks. ATAC-seq reads were counted inside union peaks using featureCounts from Rsubread (setting: isPairedEnd = T, requireBothEndsMapped = T, countChimericFragments = F, allowMultiOverlap = T) (*Liao et al., 2013*). The resulting count matrix was used as input for DESeq2 analysis (*Love et al., 2014*). Peaks were called as differentially accessible using the criteria of log2FoldChange >0.5 and padj <0.1.

## In silico sequence analysis

Comparison of DRMS DNA sequences and identification of the 17 bp motif was performed using BLAST (https://blast.ncbi.nlm.nih.gov/) and Gene Palette software (*Rebeiz and Posakony, 2004*). Analysis of the 17 bp motif was performed using Meme Suite with the TOMTOM Motif Comparison Tool (*Gupta et al., 2007*), which was run using default to compare the motif against the combined *Drosophila* databases.

## Acknowledgements

The authors would like to thank David Bilder for his gift of unpublished stocks and Stephen Pratt for statistical advice. We thank current members of the Hariharan and Harris labs for useful feedback, especially Melanie Worley, Jack StPeter, Weston Quinn and Jacob Klemm. We thank the Bloomington Stock Center, *Drosophila* Genomics Resource Center, Developmental Studies Hybridoma Bank,

and BestGene for stocks, reagents, and services. This work was funded by NIH grant R35 GM122490 and a Research Professor Award from the American Cancer Society (RP-16238–06-COUN) to IKH, and by NIH grant R35 GM128851 and a Research Scholar Award from the American Cancer Society (RSG-17-164-01-DDC) to DJM.

## Additional information

### Funding

| Funder | Grant reference number | Author |
|---|---|---|
| National Institutes of Health | R35 GM122490 | Iswar K Hariharan<br>Iswar K Hariharan |
| American Cancer Society | RP-16238-06-COUN | Iswar K Hariharan<br>Iswar K Hariharan |
| National Institutes of Health | R35 GM128851 | Daniel J McKay |
| American Cancer Society | RSG-17-164-01-DDC | Daniel J McKay |

The funders had no role in study design, data collection and interpretation, or the decision to submit the work for publication.

### Author contributions

Robin E Harris, Conceptualization, Data curation, Formal analysis, Supervision, Validation, Investigation, Visualization, Methodology, Writing - original draft, Project administration, Writing - review and editing; Michael J Stinchfield, Data curation, Investigation, Visualization, Writing - review and editing; Spencer L Nystrom, Daniel J McKay, Software, Formal analysis, Supervision, Funding acquisition, Investigation, Visualization, Writing - review and editing; Iswar K Hariharan, Conceptualization, Supervision, Funding acquisition, Writing - original draft, Project administration, Writing - review and editing

### Author ORCIDs

Robin E Harris ⓘ https://orcid.org/0000-0001-6945-6741
Michael J Stinchfield ⓘ https://orcid.org/0000-0002-6307-7270
Spencer L Nystrom ⓘ http://orcid.org/0000-0003-1000-1579
Daniel J McKay ⓘ https://orcid.org/0000-0001-8226-0604
Iswar K Hariharan ⓘ https://orcid.org/0000-0001-6505-0744

### Decision letter and Author response

Decision letter https://doi.org/10.7554/eLife.58305.sa1
Author response https://doi.org/10.7554/eLife.58305.sa2

## Additional files

### Supplementary files

• Supplementary file 1. Pairwise comparisons of ATAC-seq data showing peaks above cutoff values in between each sample: young undamaged *vs.* young damaged, old undamaged *vs.* old damaged, young undamaged *vs.* old undamaged, young damaged *vs.* old damaged. For each comparison the basic peak attributes are shown (chromosome, start and end dm3 coordinates, width in bp, unique peak ID and mean signal) and values of the peaks in each sample comparison (log2 fold change in signal, standard error, p value and adjusted p value).

• Supplementary file 2. ATAC-seq peaks categorized as damage-responsive (above cutoffs in young undamaged *vs.* young damaged), maturity-silenced (above cutoffs in young damaged *vs.* old damaged) or as damage-responsive, maturity-silenced (DRMS) peaks (at the intersection of the two). Also shown are peaks categorized as damage-responsive in old discs (above cutoffs in old

undamaged vs. old damaged) for completeness. Peak attributes and comparison values are shown as in *Supplementary file 1*.

• Supplementary file 3. Genes associated with each peak that is categorized as damage-responsive, maturity-silenced or DRMS, as in *Supplementary file 2*. For each peak the closest two genes are shown, with the chromosome, start and stop coordinates, gene name, Flybase gene ID, the distance of the peak to the gene and the number of peaks associated with that gene.

• Supplementary file 4. List of primer sequences used for cloning *Mmp1* GFP reporters (including subdivisions and *Mmp-1-US* region), DRMS enhancer regions and *in situ* probes for detecting *aspr* expression.

• Supplementary file 5. Regeneration scoring data and bar graphs of the stacked bar charts shown in main figures throughout this work. Color schemes indicating the degree of regeneration are as in main figures, Y axis is percent of total adult flies scored. Error bars are SEM.

• Transparent reporting form

## Data availability

Sequencing data have been deposited in GEO. Accession code: GSE140755. All other data generated or analyzed during this study are included in the manuscript and supporting files.

The following dataset was generated:

| Author(s) | Year | Dataset title | Dataset URL | Database and Identifier |
|---|---|---|---|---|
| Harris RE, Stinchfield MJ, Nystrom SL, McKay DJ, Hariharan IK | 2019 | Chromatin landscape changes of regenerating wing imaginal discs | https://www.ncbi.nlm.nih.gov/geo/query/acc.cgi?&acc=GSE140755 | NCBI Gene Expression Omnibus, GSE140755 |

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
