## [Decision Letter]

[Editors’ note: the authors submitted for reconsideration following the decision after peer review. What follows is the decision letter after the first round of review.]

Thank you for choosing to send your work, “Regenerative capacity in *Drosophila* imaginal discs is controlled by damage-responsive, maturity-silenced enhancers”, for consideration at *eLife*. Your initial submission has been reviewed by three peer reviewers and the evaluation has been overseen by a Senior Editor in consultation with Hugo J. Bellen as the Reviewing Editor. Although there is a clear interest in the work, the reviewers have raised important issues. For the following reasons, we are unable to consider the manuscript for publication in *eLife*.

The consensus is that the data provide little support for enhancers of changing accessibility as a genome-wide mechanism regulating regeneration. The question then becomes, is the identification of several new enhancers and of two new genes relevant to regeneration sufficient for publication by itself in *eLife*? The genome-wide data are clearly presented and explain how the genes and enhancers were identified, but they make for a long and distracting story. The reviewers suggest (and we agree) that you could make the manuscript smaller by accepting the pitfalls of whole disc ATAC-seq and provide the data in supplementary sections while the majority of the paper would focus on the experimentally confirmed regenerative genes as a more compact “Research Advance” to your 2016 paper in *eLife*.

Here is a summary of the reviews and concerns:

The paper by Harris et al. investigates how regenerative capacity decreases with time in the *Drosophila* wing disc. The Hariharan group has previously identified wingless as a gene that is induced following damage to wing discs in the early third instar but less responsive in the late third instar, when there is less capacity to regenerate, and mapped an enhancer that mediates this aspect of wingless gene expression. Because restoring *wg* expression does not restore regenerative capacity to older discs, they hypothesize that *wg* may be only one of multiple genes that are required. Here they searched genome-wide for other regulatory regions that contain adjacent damage response elements and maturity silencing elements that might identify such genes.

In the first part of the paper, Harris et al. identify and characterize an enhancer from the MMP1 gene that resembles that found at *wg*, and show that it also confers damage-responsive gene expression that loses activity with time. Based on commonalities between the MMP1 and *wg* enhancers, Harris et al. then set out to identify other such enhancers on a genome wide basis. The approach taken is to use ATAC-seq to search for chromatin that is opened in response to tissue damage in early L3 but not late L3. The approach taken is to use ATAC-seq to search for chromatin that is opened in response to tissue damage in early L3 but not late L3. This approach identified the known *wg* enhancer as maturity-silenced (MS), but was not able to detect its damage regulation (DR) in early L3. The approach did not detect the known MMP1 region at all, but did detect two other maturity-silenced MMP1 regulatory regions, one of which was validated experimentally. Given that damage-reactive chromatin accessibility does not seem as useful a criterion as expected, the authors supplement their results with published data on regenerative gene expression from another group. Combining all these data, they develop a shortlist of several genes that might be associated with DRMS regulatory elements, and test and verify 5 enhancers in transgenic flies.

To test whether any genes with DRMS enhancers are indeed required for regeneration, the authors first develop a neat assay system, called DUAL, which depends on LexA regulation for cell ablation so that Gal4/UAS can be used to knock-down candidate genes. Using DUAL, they show that CG9752, a previously uncharacterized gene encoding a secreted protein they now name *asperous*, is required for full regeneration, but surprisingly *apontic* is a negative regulator of regeneration.

Essential revisions:

The biggest difference in the data set results from developmental changes in gene expression. The regions that become less accessible are more numerous in undamaged early discs versus late discs (944 vs. 729). Automatically implicating Maturity silenced genes in regeneration is therefore misleading. Additionally, the enhancer regions that get more or less accessible in response to damage is very low in number. This may suggest a technical issue (in early discs, when rn>GAL4 expressing genes are ablated the effect is more pronounced vs. later discs. This may be due to the relative abundance of rn>GAL4 expressing genes compared to non-expressing cells.).

The underlying assumption that many enhancers in genes that play role in regeneration will only become accessible in response to damage is not supported. The opposite assumption that the genes for which the enhancer accessibility changed in response to damage are directly implicated in regeneration is unfounded too. The positive controls *wg* and *mmp1* themselves do not fit these criteria and do not meet the statistical significance cut off.

The modest agreement between the genes expressed in response to damage from Khan et al., 2017 data set and genes with enhancers containing DR and MS modules again suggest that the underlying assumption in enhancer dynamics in unsound. Only 5% of genes upregulated in damaged genes contain a DR enhancer module. If DR regions had a significant role in regulating damage response one would expect a higher overlap.

The authors then go on to analyze the other peaks that they identified with DR and MS modules. The authors suggest the presence of MS or DR motifs near genes implicated in diverse signaling pathways may indicate their expression may change in damage responsive manner (for genes in hippo pathway). They suggest that the genes in Toll pathway having MS modules may indicate that they have a new role in regeneration. The authors then suggest zelda that has MS peaks that do not meet the statistically significance cut off for DR may implicate this gene in regeneration. This part of the manuscript becomes highly speculative and should be removed in the absence of experimental evidence.

The authors tested 5 DRMS enhancers by cloning reporter genes and showed that these 5 reporters do indeed get activated to varying degrees in response to damage in early discs and get tuned down in late discs (the data is compelling for 4/5 genes, meg2 is not induced noticeably in response to damage).

The authors generated a genetic means using LexA/LexO orthogonal binary system to induce tissue damage independent of GAL4/UAS binary system (DUAL). This complicated system uses a flip-out controlled GAL4 and split LexA controlled expression of rpr or egr. What is puzzling is that by using LexA that are suppressible by Gal80ts drivers (Yagi et al., 2010) the authors could reach the same outcome in a genetically easier manner. The high number of transgenic constructs that are required for using this method will limit its usage.

The authors demonstrate the use of DUAL to knock down or overexpress genes that are implicated in regeneration (*wg*, *mmp1*, *myc*, *upd1, 2, 3* and *dome*). The regenerative potential of discs can be affected by overexpressing or knocking down these factors.

Finally, the authors tested 2 of the 5 genes for which they generated reporters and showed DRMS modules. Whereas *apt* knock down increases regenerative potential aspr knock-down decreases rate of regeneration. These data are compelling and suggest DUAL to be useful for screening for effectors in regeneration.

In summary:

1) Damage-induced changes in chromatin accessibility are less frequent than the authors expected/hoped. Please can the authors draw a clear and justified conclusion about whether they still consider this mechanism significant, or whether their data rule it out. At a minimum this would include what degrees of overlap between the ATAC-seq data and the gene expression data would have been expected by chance, and whether the 5% overlap observed is significant or suggests that accessibility is not a strong criterion for DR activity. The authors should also justify p[adj]<0.1 as the cutoff for changes in chromatin accessibility, and address whether false positives from this relaxed cutoff contribute to the limited overlap with the gene expression data. Does overlap increase with more stringent cutoffs?

2) If chromatin accessibility is still considered an important indicator of damage responsiveness, please explain why the 5% of genes overlapping with gene expression data are not followed up as a set of interesting genes, and also why they do not follow up with the 28 potential DRMS enhancers defined by chromatin accessibility criteria? If damage-induced chromatin accessibility is not a significant feature of regeneration genes, then should the focus of the analysis be the 85 genes identified as DR from gene expression analysis and MS from chromatin accessibility? With neither of these gene sets is analyzed in depth, the logic of the study as a whole is unclear.

3) It is already known that chromatin accessibility of many genes reduces over time. The authors should make clear why it should be of special significance that this is also the case for genes involved in regeneration.

4) If a mutation in a single gene (*apt*) can improve regeneration, does this contradict the overall concept that multiple genes must be affected to improve or restore regeneration at late stages? Does *esc* affect *apt* expression, and if so how is this to be interpreted? Does *apt* affect gene silencing?

---

## [Author Response]

[Editors’ note: the authors resubmitted a revised version of the paper for consideration. What follows is the authors’ response to the first round of review.]

The consensus is that the data provide little support for enhancers of changing accessibility as a genome-wide mechanism regulating regeneration. The question then becomes, is the identification of several new enhancers and of two new genes relevant to regeneration sufficient for publication by itself in eLife? The genome-wide data are clearly presented and explain how the genes and enhancers were identified, but they make for a long and distracting story. The reviewers suggest (and we agree) that you could make the manuscript smaller by accepting the pitfalls of whole disc ATAC-seq and provide the data in supplementary sections while the majority of the paper would focus on the experimentally confirmed regenerative genes as a more compact "Research Advance" to your 2016 paper in eLife.

Thank you. We agree with concerns expressed by the reviewers regarding the ATAC-seq data. As advised, we have followed your advice to prepare a more compact manuscript in the form of a Research Advance to our 2016 paper in *eLife.*

Here is a summary of the reviews and concerns:The paper by Harris et al. investigates how regenerative capacity decreases with time in the Drosophila wing disc. […] Using DUAL, they show that CG9752, a previously uncharacterized gene encoding a secreted protein they now name asperous, is required for full regeneration, but surprisingly apontic is a negative regulator of regeneration.Essential revisions:The biggest difference in the data set results from developmental changes in gene expression. The regions that become less accessible are more numerous in undamaged early discs versus late discs (944 vs. 729). Automatically implicating Maturity silenced genes in regeneration is therefore misleading. Additionally, the enhancer regions that get more or less accessible in response to damage is very low in number. This may suggest a technical issue (in early discs, when rn>GAL4 expressing genes are ablated the effect is more pronounced vs. later discs. This may be due to the relative abundance of rn>GAL4 expressing genes compared to non-expressing cells.).

We agree with the reviewers that there are a number of technical issues that make it difficult to generate clear conclusions on how chromatin changes between early and late L3 discs at a genome-wide level. The ATAC-seq experiment in this work was performed using whole discs, and as such the cells exhibiting damage-responsive changes (the blastema) only comprise a small number of the total cells analyzed. Therefore, the largest number of differences captured by this method are developmental changes arising from the surrounding undamaged tissue. Moreover, the blastema could represent a larger percentage of cells in younger discs, which could prevent some of the damage-responsive changes being detected in older discs. In the revised manuscript, we have explained the limitations of our experimental system in more explicit terms. Furthermore, as advised by the reviewers, we have moved most of the ATAC-seq data to figure supplements and indicated that we have used these data only as a guide to identify regions that could contain putative damage-responsive enhancers that are also silenced in mature discs and focused on those that we have validated experimentally.

The underlying assumption that many enhancers in genes that play role in regeneration will only become accessible in response to damage is not supported. The opposite assumption that the genes for which the enhancer accessibility changed in response to damage are directly implicated in regeneration is unfounded too. The positive controls wg and mmp1 themselves do not fit these criteria and do not meet the statistical significance cut off.

The technical issues associated with whole-disc ATAC-seq experiments make it difficult to know what is going on with respect to chromatin in the cells of the blastema. Since we have experimentally validated 6 enhancers that regulate reporter gene expression in the blastema in a way that is damage-responsive and maturity-silenced, we have refocused our manuscripts on these enhancers and their associated genes. We have refrained from making any overall conclusions about chromatin states.

The modest agreement between the genes expressed in response to damage from Khan et al., 2017 data set and genes with enhancers containing DR and MS modules again suggest that the underlying assumption in enhancer dynamics in unsound. Only 5% of genes upregulated in damaged genes contain a DR enhancer module. If DR regions had a significant role in regulating damage response one would expect a higher overlap.

In the revised manuscript, we have focused on enhancers and genes that we have studied experimentally and no longer make any claims regarding the extent of overlap between regions identified in our ATAC-seq experiments and those adjacent to genes identified by Khan et al.

The authors then go on to analyze the other peaks that they identified with DR and MS modules. The authors suggest the presence of MS or DR motifs near genes implicated in diverse signaling pathways may indicate their expression may change in damage responsive manner (for genes in hippo pathway). They suggest that the genes in Toll pathway having MS modules may indicate that they have a new role in regeneration. The authors then suggest zelda that has MS peaks that do not meet the statistically significance cut off for DR may implicate this gene in regeneration. This part of the manuscript becomes highly speculative and should be removed in the absence of experimental evidence.

This section of the manuscript has been removed.

The authors tested 5 DRMS enhancers by cloning reporter genes and showed that these 5 reporters do indeed get activated to varying degrees in response to damage in early discs and get tuned down in late discs (the data is compelling for 4/5 genes, meg2 is not induced noticeably in response to damage).

We have observed a modest damage response with meg2. However, in response to the reviewers’ comment, and because we have not conducted a more detailed study of the gene, we have removed it.

The authors generated a genetic means using LexA/LexO orthogonal binary system to induce tissue damage independent of GAL4/UAS binary system (DUAL). This complicated system uses a flip-out controlled GAL4 and split LexA controlled expression of rpr or egr. What is puzzling is that by using LexA that are suppressible by Gal80ts drivers (Yagi et al., 2010) the authors could reach the same outcome in a genetically easier manner. The high number of transgenic constructs that are required for using this method will limit its usage.

We did not explain a key difference between our system (DUAL Control) and the GAL80^ts^-based system used by others. In those studies, the Lex::GAD transcriptional regulator was used in parallel with GAL4, with the expression of both genes being regulated by GAL80. Thus, both the lexAOp-regulated and UAS-regulated transgenes are expressed simultaneously (during the ablation phase), albeit in different spatial domains. We have developed DUAL Control to specifically manipulate gene expression in the blastema and surrounding region during the regeneration phase. The heat shock that triggers ablation also activates expression of GAL4 in the surrounding tissue (by a FLP-out) which results in sustained expression in these cells during the regeneration phase. We have explained this more carefully in the revised manuscript.

The authors demonstrate the use of DUAL to knock down or overexpress genes that are implicated in regeneration (wg, mmp1, myc, upd1, 2, 3 and dome). The regenerative potential of discs can be affected by overexpressing or knocking down these factors.

The ability of DUAL Control to manipulate gene expression during the regeneration phase enabled us to demonstrate these effects convincingly.

Finally, the authors tested 2 of the 5 genes for which they generated reporters and showed DRMS modules. Whereas apt knock down increases regenerative potential aspr knock-down decreases rate of regeneration. These data are compelling and suggest DUAL to be useful for screening for effectors in regeneration.

Thank you. We agree and have now shifted the focus of the paper to these observations.

In summary:1) Damage-induced changes in chromatin accessibility are less frequent than the authors expected/hoped. Please can the authors draw a clear and justified conclusion about whether they still consider this mechanism significant, or whether their data rule it out. At a minimum this would include what degrees of overlap between the ATAC-seq data and the gene expression data would have been expected by chance, and whether the 5% overlap observed is significant or suggests that accessibility is not a strong criterion for DR activity. The authors should also justify padj<0.1 as the cutoff for changes in chromatin accessibility, and address whether false positives from this relaxed cutoff contribute to the limited overlap with the gene expression data. Does overlap increase with more stringent cutoffs?

We have acknowledged the limitations of the genome-wide approach, and that using whole discs potentially masks signals from regenerating cells. Thus, the targets we identify are likely an underestimate, and as such we cannot evaluate the degree to which this mechanism might contribute to the regulation of the regeneration program as a whole. We have clarified that the whole genome-data is not a comprehensive list of potential enhancer regions, but using the data as a guide has allowed us to identify bona fide enhancers (damage-responsive and maturity silenced) that indeed regulate genes that function in regeneration. Thus, at least some of these enhancers display chromatin accessibility changes that correlate with their activity. This mechanism is likely to be significant for some genes, but it is difficult to make clear statements regarding the number of genes that might be regulated in this way.

2) If chromatin accessibility is still considered an important indicator of damage responsiveness, please explain why the 5% of genes overlapping with gene expression data are not followed up as a set of interesting genes, and also why they do not follow up with the 28 potential DRMS enhancers defined by chromatin accessibility criteria? If damage-induced chromatin accessibility is not a significant feature of regeneration genes, then should the focus of the analysis be the 85 genes identified as DR from gene expression analysis and MS from chromatin accessibility? With neither of these gene sets is analyzed in depth, the logic of the study as a whole is unclear.

The manuscript has been reworked to no longer rely on the ATAC-seq data as a comprehensive list of damage-responsive regions. We have removed the characterization and interpretation of the data set as a whole, and instead include it only as a means to identify four new DRMS enhancers and to characterize two of the relevant genes in greater detail.

3) It is already known that chromatin accessibility of many genes reduces over time. The authors should make clear why it should be of special significance that this is also the case for genes involved in regeneration.

We have now focused on the paper on specific genes involved in regeneration that are adjacent to experimentally-validated DRMS enhancers.

4) If a mutation in a single gene (apt) can improve regeneration, does this contradict the overall concept that multiple genes must be affected to improve or restore regeneration at late stages? Does esc affect apt expression, and if so how is this to be interpreted? Does apt affect gene silencing?

If regeneration in maturing discs is indeed suppressed by the silencing of multiple DRMS enhancers that regulate genes promoting regeneration, it could be restored, at least in principle, in two different ways: (1) By alleviating the silencing at multiple DRMS enhancers – knockdown of *esc* likely functions in this way, and (2) by activating pathways that function downstream of genes that promote regeneration (bypass suppression). Our previous work (Worley et al., 2018, *eLife*) has shown that the JAK/STAT pathway can promote plasticity during regeneration. Knockdown of *apt* increases expression of the STAT reporter. Perhaps this still poorly-understood mechanism acts in parallel to the alleviation of PcG-mediated silencing.

We have shown that *esc* knockdown does de-repress the *apt* enhancer. However, the antibody staining was not sufficiently robust to determine whether Apt protein levels were also affected.

*esc* knockdown likely depresses many maturity-silenced enhancers and those that we have studied are probably a small subset. Since the effect of *esc* knockdown is to improve regeneration, it is likely that it the effect of derepressing genes that promote regeneration must outweigh the effect of derepressing those that restrict regeneration. The genes that we have characterized that are regulated by DRMS enhancers (*wg*, *Wnt6*, *Mmp1*, *asperous/CG9572* and *apontic*) represent four positive regulators of regeneration and one negative regulator of regeneration. It is often the case that the induction of specific cellular programs include the induction of inhibitors of that same program. We think that *apt* falls into that category.

We have modified the Discussion to address these points.